# The Need for Fission Track Data Transparency and Sharing

Murat T. Tamer[1,2], Ling Chung[3], Richard A. Ketcham[2], Andrew J. W. Gleadow[3]

[1]State Key Laboratory of Earthquake Dynamics, Institute of Geology, China Earthquake Administration, Beijing 100029, China
[2]Jackson School of Geosciences, The University of Texas at Austin, 78712 Austin, Texas, U.S.A.
[3]School of Geography, Earth and Atmospheric Sciences, University of Melbourne, 3010 Melbourne, Victoria, Australia

*Correspondence to*: Murat T. Tamer (murat@ies.ac.cn)

**Abstract.** We report a new image-based inter-analyst study to investigate apatite fission-track grain selection and analysis by 13 participants from an image data set that included grains of variable quality. Results suggest that participants with less experience selected a higher percentage of unsuitable grains, while participants from the same laboratories generally provided similar results. Less analytical experience may result in the rejection of suitable grains, or inclusion of unsuitable ones. While inappropriate omission and inclusion can both bias results, the latter is more pernicious due to the standard practice of achieving a predecided number of analyses; particularly in difficult samples, there is a danger of "compromising data quality" and integrity by weakening selection criteria. Juxtaposing selected regions of interest (ROIs) on the same grains indicates that zoned grains and grains with inclusions and defects yield varying track density estimates, indicating that ROI placement can be an influential factor. We hereby propose the development of image data repositories for the purpose of achieving global data transparency. In this repository, images and analyses can be accessed, reviewed, and re-analysed. In addition, we propose the implementation of a global guidance for fission-track analysis, digital teaching modules, and open science. We also point out the need for new approaches to zeta calibration that include consideration of grain quality, methods of uranium determination, and etching protocols.

## 1 Introduction

Apatite fission-track dating and thermal history modeling are widely used for near-surface research in earth sciences, across a large spectrum of subjects such as landscape evolution (Reiners and Shuster 2009; Lemot et al., 2023; Gallen et al., 2023), climate change (Barnes et al., 2012; Qiu and Liu 2018; Yu et al., 2022), glacier-driven exhumation (Balestrieri et al., 1991; Fitzgerald and Goodge 2022; Karaoğlan et al., 2023), natural resource exploration (Dumitru et al., 1991; Deng et al., 2015; Qiu et al., 2023; Gülyüz et al., 2024) and biodiversity (Kohn et al., 1992; Torres et al., 2013; Bernet et al., 2023). With the prerequisite that suitable apatite crystals are available, six essential 'ingredients' are required for fission-track time-temperature modelling, the first (1) being selection of grains on the polished and etched grain mount that are suitable for analysis, which then consists of the track densities calculated from (2) track counts over (3) a selected region of interest; (4) preferably more than a few tens of confined track lengths per sample; (5) mean etch figure diameter parallel to c-axis ($D_{par}$) (Donelick, 1993;

Burtner et al., 1994; Donelick et al., 1999), or chemical information to infer kinetics; and (6) an estimate of the $^{238}U$
concentration (Tagami and O'Sullivan 2005). All of these inputs are still largely analyst-driven, although some new
technologies are being developed to alleviate this. Recent developments in image analysis and AI have contributed significant
advances in auto-counting and auto-measurement (Gleadow et al., 2009, 2019; Nachtergaele and De Grave 2021; Li et al.,
2022; Ren et al., 2023; Boone et al., 2023b), but are not yet in a position to replace human decision making. Similarly, laser
ablation mass spectrometry has become an alternative (Hasebe et al., 2004) to the widely used external detector method (EDM)
(Gleadow and Lovering 1977) for uranium content determination, and U mapping has been developed to help account for U
zonation (Ansberque et al., 2021); these obviate some human decisions, but may add others.

Previous apatite fission-track inter-laboratory and inter-analyst experiments showed significant variation in measurements for
the same samples and even standards (Naeser et al., 1981; Miller et al., 1985, 1990, 1993; Barbarand et al., 2003; Ketcham et
al., 2009; Sobel and Seward, 2010; Ketcham et al., 2015; Ketcham et al., 2018; Tamer et al., 2019). These variations have been
attributed to a broad range of factors, including instrumentation, analytical preferences, etching protocol, and analyst selection
criteria. A common feature in these experiments (except Tamer et al., 2019, which compared only two analysts) is that there
was no direct control over what analysts observed. In most cases participants had their own aliquots of study samples, and in
experiments where all analysts measured the same grain mounts they undoubtedly looked at different sets of grains. Moreover,
until the advent of efficient computational tools, there has been a limited ability to document and compare the counted areas
within measured grains. As a result, ingredients (1) and (3) above have not been quantitatively explored as sources of variation
in dates, even though they may exert a first-order influence on the data quality and extractable thermal history information.
Similarly, again with the exception of Tamer et al. (2019), different analysts have never evaluated and measured the same sets
of features for confined track measurement, and their decisions concerning individual features have not been captured, limiting
the means to compare and evaluate ingredient (4).

From a given set of grains, grain selection influences results in several ways. Grains where oily and aqueous fluids have
penetrated into tracks may hinder the recognition of some surface tracks, or cause confined tracks to appear shorter and thus
more annealed (Ketcham et al., 2015), leading respectively to underestimation of ages or overestimation of temperatures.
Grains with excessive defects, such as polishing artefacts or etched dislocations, may cause misidentification of some spurious
features as actual tracks and cause overestimation of ages. Conversely, recognition of some defects in a sample may cause an
analyst to alter their counting criteria to be more likely to reject borderline features they interpreted as tracks when conducting
their zeta calibration measurements. Track density can vary by up to 35% if the grain is not oriented with the c-axis in the
viewing plane (Aslanian et al., 2022), making both the resulting age and etch figure dimensions (e.g. $D_{par}$) inaccurate, thereby
affecting estimates of kinetics and initial track length. A perceived need to meet targets for the number of grains analysed may

cause an analyst to select borderline-suitable grains or tracks that may not have been selected otherwise (Tamer and Ketcham, 2023).

Whereas the area counted for fission-track density determinations has historically been defined by squares in an eyepiece reticule, recent image-based systems allow the user to draw an arbitrarily shaped region of interest. In both cases, this process must be executed with care. Regions of interest need to be placed so that the grain surface they encompass is not biased with respect to the ability to host detectable tracks. Regions of interest within one fission-fragment range of the grain edge will not sample tracks from a full $4\pi$ geometry (Fleischer et al., 1975) and including sizable defects and cracks in the region of interest may result in uncountable areas; both effects will bias ages towards lower values. Regions of interest that include zones with different U content complicate the accurate determination of U across the track-generating region (Vermeesch, 2017), and suffer edge effects from sampling a $4\pi$ region that hosts variable U concentration. This bias can result both from methods using laser ablation, where typically a smaller area is sampled for the U-determination than for the spontaneous track count, and the external detector method, where perfect matching between spontaneous and induced track regions of interest can be difficult to achieve, especially where the track density is low.

The minimum recommended number of grains for age measurements for igneous-type samples is ~10 (Wagner and Van den Haute, 1992), and at least 20 for sedimentary rocks (Kohn et al., 2024). In both cases, even more are needed if there is any indication of kinetic variation (Donelick et al. 2005). For detrital samples the recommended number is ~120 or more (Vermeesch 2004). These guidelines are enforced not only at the laboratory level, but during the peer review process as well. If grains are few or of low quality, an analyst may consciously or unconsciously add some borderline-quality grains to meet goals for data quantity. Similarly, having pre-determined goals for the number of confined tracks per sample can incentivize accepting lengths that might otherwise be passed over. For both data types, aiming for specific quantities of data may eventually cause a loss of quality.

We carried out a new apatite fission track inter-analyst experiment designed to investigate variability in grain selection and region of interest definition criteria. Building upon a previous two-analyst study (Tamer et al., 2019), we also tested the identification of confined track lengths. Participants were asked to perform apatite FT analysis on a selection of grains drawn from an identical image set featuring variable grain quality using software that records all details of the analysis as overlays in an .xml file, thereby allowing for subsequent review. Analysts were also asked to fill out a questionnaire about their approach.

**2 Materials and Methods**

## 2.1 Image Data Repository

We created an image data repository consisting of 41 apatite grain and 3 graticule (length calibration grid on a microscope slide) images from the in-house fission-track data repositories at the University of Melbourne (UM) and the University of Texas at Austin (UT). Images from UM were captured by Ling Chung using a Zeiss Axio Imager M1m microscope with an IDS µEye camera and white balance correction, while images from UT were taken by Sean Sanguinito and Murat Tamer,

| Grain | Image Source | Description |
|---|---|---|
| 1 | UM | Not parallel to c-axis, not suitable |
| 2 | UM | Fluid in tracks, not suitable |
| 3 | UM | Not 100% C-parallel and with low number of inclusion but suitable* |
| 4 | UM | Suitable grain. |
| 5 | UM | Suitable if the parts with inclusions are excluded |
| 6 | UM | Not parallel to c-axis, not suitable |
| 7 | UM | Suitable if the parts with inclusions and dislocations (left hand corner) are excluded |
| 8 | UM | Too many inclusions, not suitable |
| 9 | UM | Suitable if the parts with inclusions and cluster of small disturbing surface features are excluded |
| 10 | UM | Not parallel to c-axis, not suitable |
| 11 | UM | Exclusively for length measurement |
| 12 | UM | Exclusively for length measurement |
| 13 | UM | Exclusively for length measurement |
| 14 | UT | Exclusively for length measurement |
| 15 | UT | Exclusively for length measurement |
| 16 | UM | Suitable if the parts with inclusions and dislocations (left hand corner) are excluded. |
| 17 | UM | Suitable if the parts with inclusions are excluded |
| 18 | UM | Suitable if the parts with inclusions are excluded |
| 19 | UM | Suitable if the parts with inclusions are excluded |
| 20 | UM | Suitable grain |
| 21 | UM | Suitable grain |
| 22 | UM | Suitable grain * |
| 23 | UM | Not parallel to c-axis, not suitable |
| 24 | UT | Fluid in tracks and noticeable uneven track distribution, not suitable |
| 25 | UM | Suitable grain |
| 26 | UM | Suitable grain* |
| 27 | UM | Suitable grain* |
| 28 | UM | Suitable grain* |
| 29 | UM | Suitable grain |
| 30 | UM | Suitable grain if the parts with inclusions and dislocations are excluded. |
| 31 | UT | Too many inclusions, not suitable |
| 32 | UT | Not 100% C-parallel and noticeably uneven track distribution. Borderline grain* |
| 33 | UT | Not parallel to c-axis, obvious uneven track distribution, not suitable |
| 34 | UT | Too many inclusions, not suitable |
| 35 | UT | Borderline grain |
| 36 | UT | Too many inclusions, not suitable |
| 37 | UT | Low track density, be careful with region of interest selection. Suitable grain |
| 38 | UT | Too many inclusions, not suitable |
| 39 | UT | Not 100% C-parallel but suitable |

| 40 | UT | Too many inclusions, not suitable |
| 41 | UT | Suitable grain |
| 42 | UM | 50x2 micron graticule |
| 43 | UT | Pyser-SGI Graticule 02A00429 S16 Stage MIC 1mm/0.01mm |
| 44 | UT | Pyser-SGI Graticule 02A00429 S16 Stage MIC 1mm/0.01mm |

**Table 1: Description of images. UT: University of Texas at Austin, UM: University of Melbourne. *: If U ppm is determined using LA-ICP-MS approach, need to cross check counting area as track distribution is slightly uneven.**

using a Zeiss Axio Imager M2m microscope with Olympus SYS UC30 camera and no white balance. Grains from UT were etched with 5.5M $HNO_3$ at 21°C for 20s (Carlson et al., 1999), while the grains from UM were etched with 5M $HNO_3$ at 20°C for 20s (Gleadow et al. 1986; Green et al., 1986). The images used in this study can be viewed at geochron@home (Vermeesch 2024, https://doi.org/10.5281/zenodo.13777917). 36 grains were selected for track density measurements and 5 for confined track length measurements. To test the self-reproducibility of the analytical results, we repeated one grain image as two

different grain areas (Grains 07 and 16). The grain descriptions are given in Table 1.

## 2.2 Announcement and Participant Instructions

The announcement of the study was made at the 17th International Conference on Thermochronology, 2021, Santa Fe (Tamer et al, 2021) and in relevant email lists. The participants were asked to perform track density and confined track length and $D_{par}$ measurements using their preferred approach, including any analytical software, manual measurement, or AI-based analysis.

The participants were not instructed to reach a given number of grains or confined track length analyses but were instructed to skip or accept grains for analysis according to their own judgement.

This experiment utilized Fission Track Studio, a dual software suite developed by the Melbourne Thermochronological Research Group (MTRG) that is capable of automatic grain stack-image acquisition (TrackWorks) and image review and

120 measurement (FastTracks). The FastTracks program offers manual and automated analytical tools for obtaining all essential parameters for FT dating as well as a cross-section tool for precise dip angle determination for length measurements. All analytical results were recorded in an .xml file that can be reloaded for a follow-up analysis and review. The University of Melbourne provided a temporary FastTracks license and a detailed user manual for those who wanted to participate in this study. The participants had the option to reveal their names and affiliations or to be anonymous. A participant's submission

was accepted only if the analysis was performed by a single analyst.

## 2.3 Reviewer Criteria

In the absence of absolute standards, we used the grain selection criteria of L. Chung and the confined track length measurement judgments of M. Tamer as reference points for the review of the participant results. However, no fission-track analyst can claim complete certainty in their judgments about track features and we do not suggest that these reference results

represent 'true' values. Rather they are simply used as reference values that are probably typical of reasonably experienced analysts. They were used as the starting point for a detailed grain-by-grain and track-by-track discussion with the participants

to arrive at a consensus view and to ascertain which factors are most likely to lead to discrepancies between analysts. Such a detailed analysis has not previously been undertaken to our knowledge and would be all but impossible without the image-based approach used here. To test the objectivity of the reviewers, the participants were shown those of their selections and measurements that the reviewers considered questionable or in some cases, erroneous, after which they could acknowledge or dispute the review.

Grains were judged to be suitable, unsuitable, or borderline by L. Chung. A grain having any of the three following properties was considered unsuitable: (1) the polished surface was not parallel to the c-axis (Fig. 1 a, b), (2) fluids were present in tracks (Fig. 1 c, d), (3) an excessive number of inclusions/defects/uncertain features was intermingled with actual tracks (Fig. 1 e, f). Additionally, heterogeneous U distribution within the grain, judging from the distribution of spontaneous tracks, can be a complicating factor, especially if LAICPMS spot analysis is used for U determination. Induced tracks in the EDM detector can provide an improved basis for recognizing zoning, but zoning can also accentuate the consequences from misalignment of the spontaneous and induced track regions. In samples with low abundance and/or low-quality of grains, some borderline-quality grains which are not %100 parallel to c-axis and/or contain distinguishable abundance of defects and inclusions may be included in the resulting data sets (Fig. 1 g, h). Selection of the region of interest may become a challenge for inclusion/defect-rich (Fig 1 i, j) and zoned grains (Fig 1 k, l). A track is defined as "confined" if both ends are not exposed at the surface (Fig. 1 m, n), and the length of the confined track is measured as long as the track tips are visible and unobscured by surrounding features (Fig. 1 o, p). Moreover, it is important that confined tracks are not filled, fully or partially, with fluid, such as may result from fingerprints, which significantly affects their optical contrast within the host mineral (Fig 1 q, r). Any measurement that does not meet these criteria is considered invalid.

Thirteen analysts returned completed questionnaires, though one of them could not provide the .xml file due to technical problems. Although L. Chung and M. Tamer evaluated each other's analyses, we consider them as reviewers in this study. Two additional analysts from the same laboratory with different years of experience submitted answers in one combined .xml file. Because they were unable to disentangle their results or reconduct them independently, their results are not included in the analysis below. While some participants wished to remain anonymous, others chose to be transparent with their identities; Table S1 provides the list of participants. Participants' overall and recent experience and their current fission-track setup, a summary of the questionnaire, selection percentages for grains, and validity percentages for confined track lengths are provided in Table 2. Excluding one of the repeated grains for checking self-reproducibility (Grain 16), of the 35 grain image sets, L. Chung estimated that 22 are suitable and 13 are unsuitable for fission track analysis. In grain-by-grain checking, we counted how many of the suitable grains were selected and how many of the unsuitable grains were rejected. Some of the participants used FastTracks' automatic tools for c-axis orientation and $D_{par}$ length measurements, but we did not track whether these results were accepted as-is or subsequently modified. The resulting suitable and unsuitable grain selection percentages are reported as percentages. M. Tamer examined every measured confined track length to check validity as determined above.

The percentage of valid measurements is reported as the percentage of measured tracks divided by the total number of valid measurements. We did not evaluate how many valid tracks were excluded, as there was no way to determine whether such tracks were intentionally omitted or simply missed.

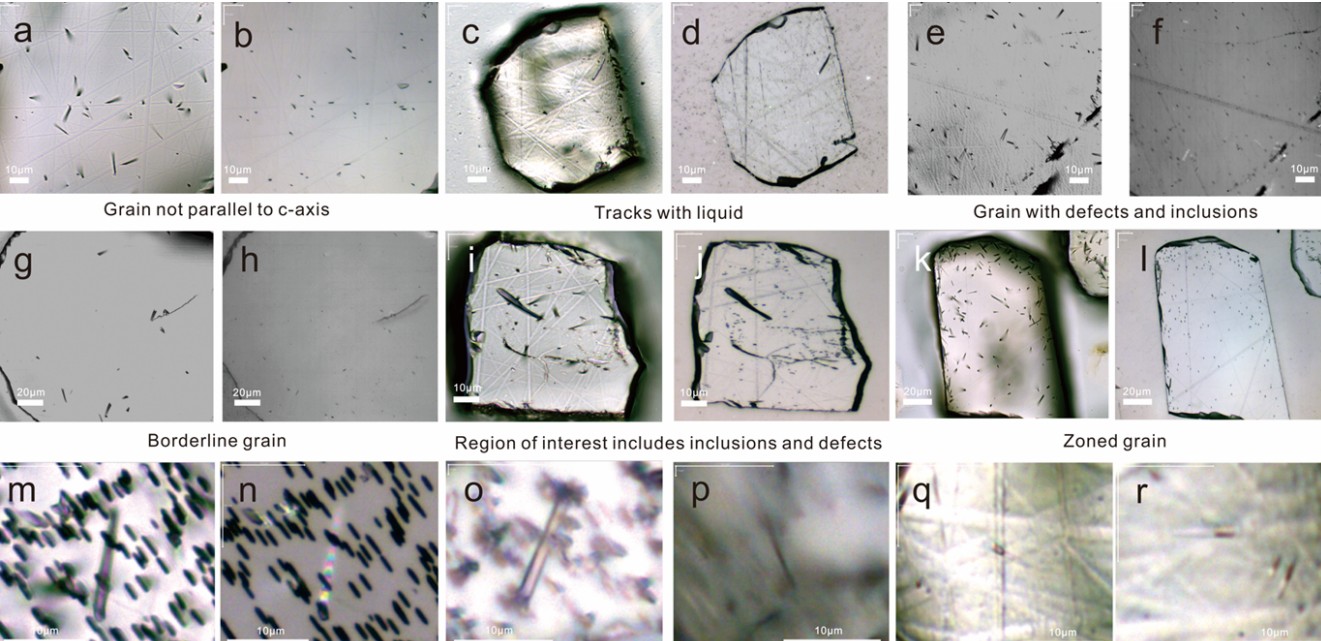

Figure 1: Examples of grains and tracks that could cause erroneous grain and region of interest selection or invalid confined track length measurement for fission-track analyses. (a) A grain not parallel to the c-axis appears to be suitable in transmitted light, (b) but varying etch pit orientations clearly show that the grain is not parallel to the c-axis in reflected light. (c,d) Transmitted and reflected light images showing how a dirty grain surface with liquids in tracks can impair track recognition. (e,f) Transmitted and reflected light images of a grain showing an excess of non-track features such as defect and inclusion that are difficult to distinguish from actual tracks. (g,h)Transmitted and reflected light images of a grain with a low and uneven track density, suggesting the possible presence of zoning that would need to be accounted for. (i,j)Transmitted and reflected light images of a grain showing large defects and inclusions occupying space obscuring fission tracks, resulting in underestimation of the fission-track density if these features are included in the region of interest. (k,l) Transmitted and reflected light images of a zoned grain, for which different placements of the region of interest and location of ablation point(s) may yield divergent ages. (m,n)Transmitted and reflected light images of a would-be confined track, where the reflected light image shows one tip may be exposed to the surface. (o,p) Confined tracks may also be rendered invalid for measurement by obscuring features, (q,r) or partial fluid fillings.

| Analyst | Total years of experience | Activity in the past two years | In-house fission-track setup | Auto c-axis assignment tool | Graticule measurement |
|---|---|---|---|---|---|
| 1 | 7 | Yes | AS | Yes | No |
| 2 | 14 | Yes | AS | No | Yes |
| 3 | 7 | Yes | AS | No | Yes |
| 4 | 6 | Yes | AS | Yes | No |
| 5 | 5 | Yes | AS | Yes | No |
| 6 | 4 | Yes | C | Yes | No |
| 7 | 6 | Yes | C | No | No |
| 8 | 4 | Yes | AS | No | No |
| 9 | 5 | Yes | AS | No | No |
| 10 | 17 | Yes | AS | Yes | Yes |
| 11 | 30 | No | D | No | Yes |
| 12 | 2 | Yes | AS | No | No |
| 13 | 5 | Yes | AS | No | No |
| 14 | 9 | Yes | AS | No | Yes |
| 15 | 2 | Yes | AS | No | No |
| 16* | 40 | Yes | C | N/A | N/A |

**Table 2: Summary of the questionnaire, percentages of grain selections, and confined track length measurement validities.**

**AS: Autoscan, C: Custom; D: Dumitru System. *: The analyst did not participate in the experiment but evaluated the grains as suitable and unsuitable.**

## 3 Results and Discussion

### 3.1 Graticule Calibration

Graticule images taken by microscopes at UM and UT were included in the data set for calibration. Although calibration is an essential step before performing an analysis, only five participants reported measuring them. Some omissions may have been due to not fully understanding the terms of the experiment. To make the comparison of results easier, we used the default graticule calibration for all analysts. The graticule measurements are summarised in Table S3. Using default calibration,

analysts performed measurements with >99.0% accuracy. Considering the limits of optical microscopy, this accuracy provides measurements within analytical errors.

## 3.2 Self-Reproducibility

Grains 7 and 16 are duplicated images of the same grain in our data set. While some participants skipped Grain 16 after noticing the repetition, some performed density measurements on both grains. Although these remeasurements demonstrated high self-reproducibility (Fig 2), minimum and maximum densities vary by ±30%. The difference can be traced to the varying region of interest selection, light source utilization (transmitted only, reflected only or both), and track counting routines. Although the zeta method (Hurford and Green, 1983) is intended to normalize some differences among analysts, the degree of variation calls into question whether the normalization approach is being asked to accomplish more than it should.

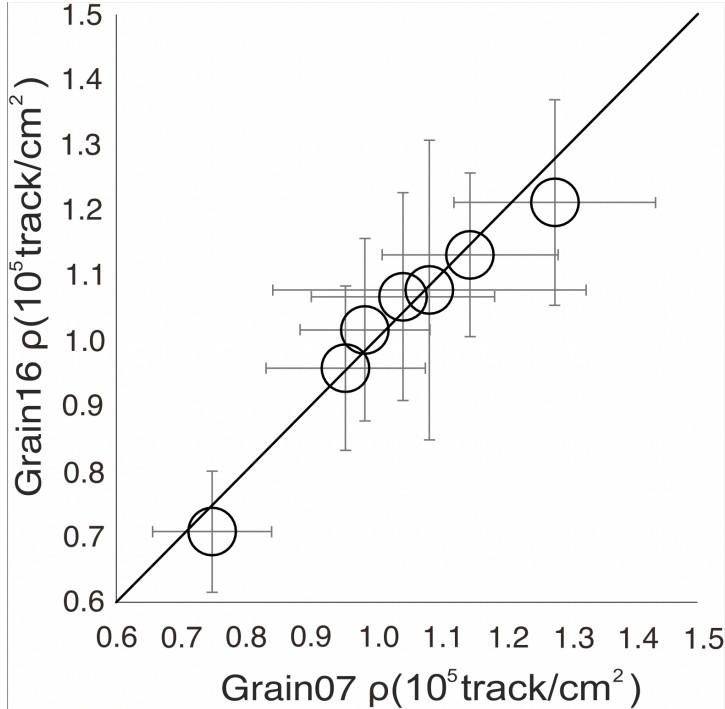

**Figure 2: Self-reproducibility of track density (ρ) determinations on replicated grain images for seven analysts.**

## 3.3 Post-review follow-up and objectivity of the review

After the initial review of grain and confined track length measurements, a follow-up meeting with each participant was conducted to discuss each judgment deemed questionable or unsuitable by the reviewers. Virtually all participants acknowledged all inappropriate grain selections and confined track length measurements, except Participant 6 considered Grain 08 and Grain 31 as borderline instead of unsuitable. This high rate of acknowledgment by the participants supports the

soundness of the criteria utilized by the reviewers. According to the participants, inappropriate selection and measurements stemmed from different factors. While some participants cited a lack of attention to details (e.g. poor identification of track ends), others stated that they have been choosing some unsuitable grains in their routine fission-track studies since their training. Some of the participants mentioned that they knowingly added unsuitable grains to the data sets in the past to meet the expected number of grains.

### 3.4 Track Density and Confined Track Length Distributions

This study is designed principally to evaluate identification of individual features rather than measurement averages and standard deviations. If invalid track lengths are measured or unsuitable grains are analyzed which give results similar to valid lengths or suitable grains, summary statistics will not suggest any problem; in fact, they would appear to improve by raising the number of analyses, the perverse incentive we wish to counteract. We calculated the suitable and unsuitable grain selection percentages (Table 3) based on the number of each grain type as listed in Table 1. The confined track length measurement validity percentage is calculated as the number of valid track length measurements divided by the total number of confined track length measurements for each analyst (Table 4). There are several valid lengths etched by both 5.0M and 5.5M etchants that may be under-etched, which were not measured by the reviewer but were measured by some analysts. The reviewer evaluated the lengths based only on the criteria laid out in section 2.3, and not specifically by how well-etched the tracks are..

When compiling summary statistics, we separated the density estimates based on suitable grains and valid track length measurements from estimates the included unsuitable ones, and compared them using dispersion and $\chi^2$ probability values for the density data and the mean and standard deviation of mean lengths (Table 3 and 4). Initial density determinations yield a dispersion value of 10 and a $\chi^2$ probability of 0.03. Exclusion of unsuitable grain data provided significant improvements in net density similarity, with dispersion and $\chi^2$ probability respectively 0 and 0.42 (Table 3). Excluding invalid confined tracks raises the average mean length by ~0.3 μm, well beyond the 0.08 μm precision limit estimated by the doubling the standard error, and reduces the group standard deviation by 20% (Table 4).

The histograms of track density and confined track length distributions of each participant provide additional insights (Fig 3). The density distributions of suitable grains are more consistent than for unsuitable grains, and the inclusion of unsuitable grains in all cases skewed the track density distribution to lower values. Participants 1 and 10 and Participants 8 and 9 are from the same two laboratories and their results similarities in their respective track density results. This may be related to the shared training and/or analytical routine in counting, though for length measurements participants 8 and 9 had more divergent results, possibly due to different personal selection criteria. The confined length histograms indicate that both the number and the choice of tracks measured varied considerably between participants. $D_{par}$ measurements of suitable grains tend to be more similar with some outliers, while the $D_{par}$s of unsuitable grains have higher dispersion (Fig S1). Post-experiment interviews

with participants suggested that the dispersion of $D_{par}$s on suitable grains may have stemmed from different levels of zoom applied.

| Analyst | All reported density measurements | | | Density measurements on suitable grains | | | Density measurements on unsuitable grains | | | Suitable grain selection percentage (%) | Unsuitable grain selection percentage (%) |
|---|---|---|---|---|---|---|---|---|---|---|---|
| | N | $\rho$ ($10^5$ track/cm$^2$) | $\sigma$ ($10^5$ track/cm$^2$) | N | $\rho$ ($10^5$ track/cm$^2$) | $\sigma$ ($10^5$ track/cm$^2$) | N | $\rho$ ($10^5$ track/cm$^2$) | $\sigma$ ($10^5$ track/cm$^2$) | | |
| 1 | 22 | 6.25 (35) | 3.42 | 21 | 6.47 (73) | 3.34 | 1 | 1.63 | N/A | 95 | 8 |
| 2 | 22 | 6.25 (82) | 3.84 | 22 | 6.25 (82) | 3.84 | 0 | N/A | N/A | 100 | 0 |
| 3 | 29 | 4.81 (60) | 3.23 | 21 | 5.62 (70) | 3.21 | 8 | 2.68 (80) | 2.25 | 95 | 62 |
| 4 | 23 | 5.63 (72) | 3.44 | 20 | 5.82 (59) | 3.65 | 3 | 4.36 (46) | 0.79 | 91 | 23 |
| 5 | 11 | 8.34 (114) | 3.79 | 11 | 8.34 (114) | 3.79 | 0 | N/A | N/A | 50 | 0 |
| 6 | 26 | 6.71 (70) | 3.57 | 21 | 6.97 (78) | 3.56 | 5 | 5.62 (170) | 3.80 | 95 | 38 |
| 7 | 26 | 5.95 (76) | 3.86 | 22 | 6.60 (82) | 3.84 | 4 | 2.42 (60) | 1.19 | 100 | 31 |
| 8 | 35 | 5.11 (60) | 3.55 | 22 | 6.23 (85) | 3.97 | 13 | 3.22 (39) | 1.41 | 100 | 100 |
| 9 | 34 | 4.92 (61) | 3.53 | 22 | 6.05 (81) | 3.78 | 12 | 2.86 (49) | 1.70 | 100 | 92 |
| 10 | 22 | 6.21 (83) | 3.89 | Reviewer | | | Reviewer | | | Reviewer | Reviewer |
| 11 | 21 | 6.25 (45) | 1.81 | 21 | 6.38 (71) | 3.26 | 1 | 3.45 | 3.24 | 95 | 8 |
| 12 | 31 | 6.14 (67) | 3.75 | 21 | 6.87 (84) | 3.86 | 10 | 4.62 (99) | 3.13 | 95 | 77 |
| 13 | 28 | 4.22 (47) | 2.51 | 21 | 4.85 (56) | 2.57 | 7 | 2.32 (31) | 0.83 | 95 | 54 |
| 14 | 24 | 5.76 (70) | 3.44 | 22 | 5.91 (74) | 3.46 | 2 | 4.36 (274) | 3.87 | 100 | 15 |
| 15 | 22 | 6.75 (73) | 3.41 | 19 | 7.31 (76) | 3.32 | 3 | 3.20 (59) | 1.03 | 86 | 23 |
| 16* | 22 | N/A | N/A | 22 | N/A | N/A | 0 | N/A | N/A | 100 | 0 |
| Dispersion | 10 | | | 0 | | | | | | | |
| $\chi^2$ | 0.03 | | | 0.42 | | | | | | | |

**Table 3: Track density estimations. N= number analyzed grains; $\rho$ = track density. *: The analyst did not participate in the experiment but evaluated the grains as suitable and unsuitable. Numbers in parentheses denote standard errors. Dispersion and $\chi2$ are calculated using Radialplotter (Vermeesch 2009). Suitable and unsuitable grain selection percentages are calculated based on the number of grains of each type selected as listed in Table 1.**

| Analyst | All reported length measurements | | | Valid length measurements | | | Invalid length measurements | | | Confined track length measurement validity percentage (%) |
|---|---|---|---|---|---|---|---|---|---|---|
| | N | $l_m$ (μm) | σ (μm) | N | $l_m$ (μm) | σ (μm) | N | $l_m$ (μm) | σ (μm) | |
| 1 | 16 | 13.91 (35) | 1.38 | 12 | 14.33 (29) | 1.00 | 4 | 12.66 (88) | 1.76 | 75 |
| 2 | 14 | 13.71 (41) | 1.54 | | Reviewer | | | Reviewer | | Reviewer |
| 3 | 31 | 11.96 (39) | 2.18 | 13 | 13.05 (47) | 1.70 | 18 | 11.17 (51) | 2.18 | 42 |
| 4 | 8 | 13.79 (51) | 1.45 | 7 | 13.82 (59) | 1.56 | 1 | 13.58 | N/A | 88 |
| 5 | 19 | 13.11 (36) | 1.59 | 13 | 13.58 (39) | 1.40 | 6 | 12.07 (64) | 1.58 | 68 |
| 6 | 5 | 13.08 (51) | 1.14 | 4 | 12.80 (55) | 1.10 | 1 | 14.22 | N/A | 80 |
| 7 | 10 | 12.43 (49) | 1.54 | 6 | 12.85 (50) | 1.23 | 4 | 11.80 (96) | 1.91 | 60 |
| 8 | 16 | 12.69 (47) | 1.90 | 9 | 12.67 (62) | 1.86 | 7 | 12.72 (79) | 2.09 | 56 |
| 9 | 33 | 13.26 (24) | 1.40 | 18 | 12.96 (33) | 1.40 | 15 | 13.63 (35) | 1.36 | 55 |
| 10 | 18 | 14.39 (36) | 1.54 | 17 | 14.42 (38) | 1.58 | 1 | 13.89 | N/A | 94 |
| 11 | 16 | 13.16 (45) | 1.81 | 13 | 13.51 (45) | 1.61 | 3 | 11.66 (125) | 2.17 | 81 |
| 12 | 4 | 12.32 (157) | 3.15 | 1 | 12.88 | N/A | 3 | 12.13 (221) | 3.83 | 25 |
| 13 | 49 | 12.05 (33) | 2.32 | 20 | 13.16 (30) | 1.33 | 29 | 11.29 (48) | 2.57 | 41 |
| 14 | 18 | 13.19 (39) | 1.66 | 14 | 13.36 (47) | 1.77 | 4 | 12.60 (62) | 1.24 | 78 |
| 15 | 22 | 12.88 (43) | 2.01 | 13 | 12.98 (34) | 1.22 | 3 | 12.73 (96) | 2.89 | 59 |

| | | | |
|---|---|---|---|
| Mean $l_m$ (μm) | 13.02 (04) | | 13.31 (04) |
| σ (μm) | 0.68 | | 0.54 |

**Table 4: Confined track length measurements. N=number of lengths; $l_m$= mean track length; σ= standard deviation. Numbers in parentheses denote standard errors. The confined track length measurement validity percentage is calculated as the number of valid track length measurements divided by the total number of confined track length measurements for each analyst.**

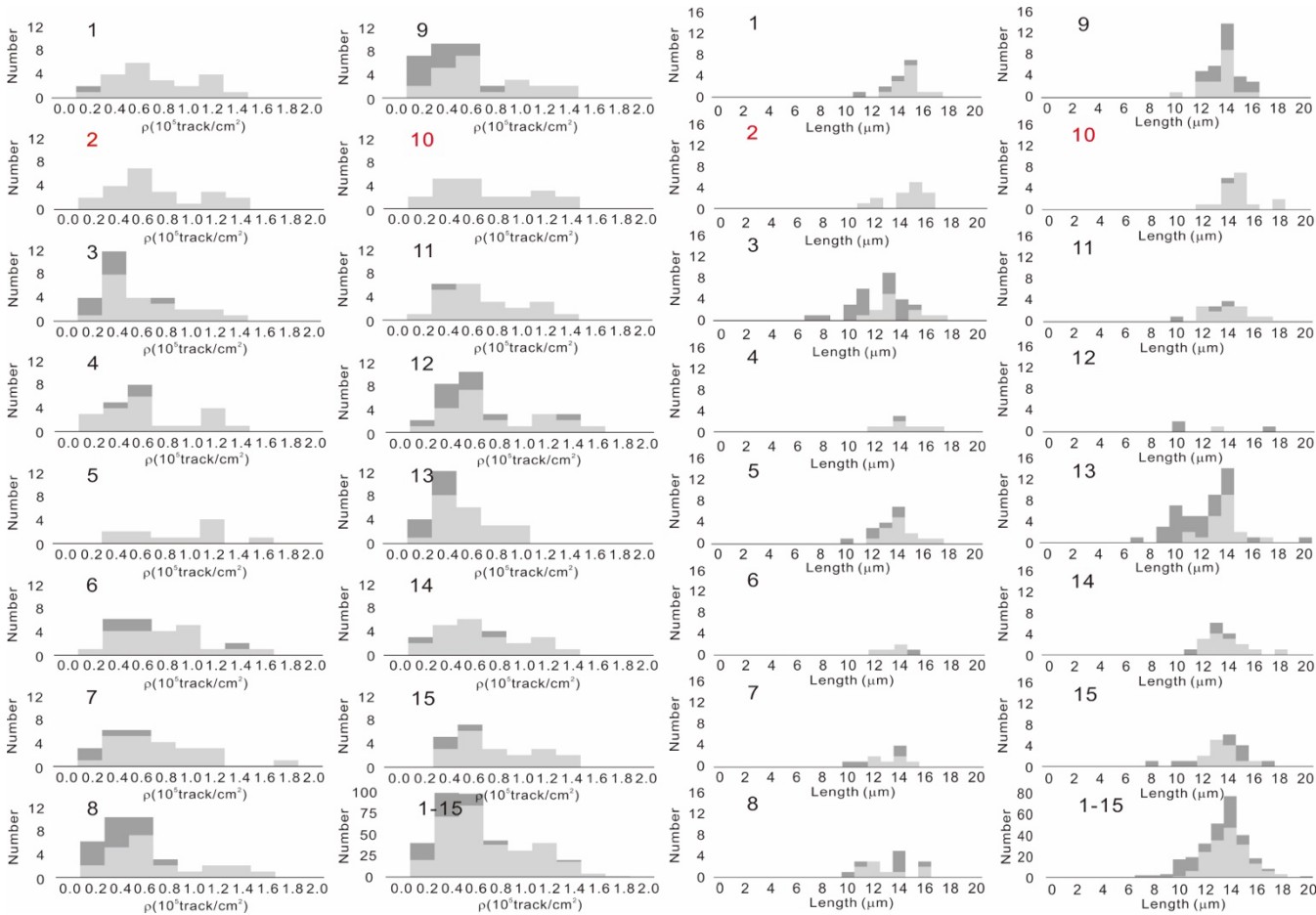

**Figure 3: Track density (ρ) and confined track length distributions of each participant and the reviewers (2 and 10, in red). Participant numbers are indicated at the top left. The cumulative result for all participants is shown at the bottom right (1-15). Dark grey shows measurements of grains and confined tracks assessed to be unsuitable, and light grey displays the measurements of suitable selections.**

## 3.5 Impact of Experience

The acceptance percentage of suitable grains shows only a weak relationship with years of experience (Fig 4a). With increasing years of experience, the acceptance percentage of unsuitable grains decreases sharply, while the confined track validity percentage increases (Fig 4b). Accordingly, those who select higher percentages of unsuitable grains tend to have lower validity percentages of confined track length measurements as well (Fig 4c). These results highlight analyst experience as an important factor in data quality, although some less experienced analysts performed as well as much more experienced ones.

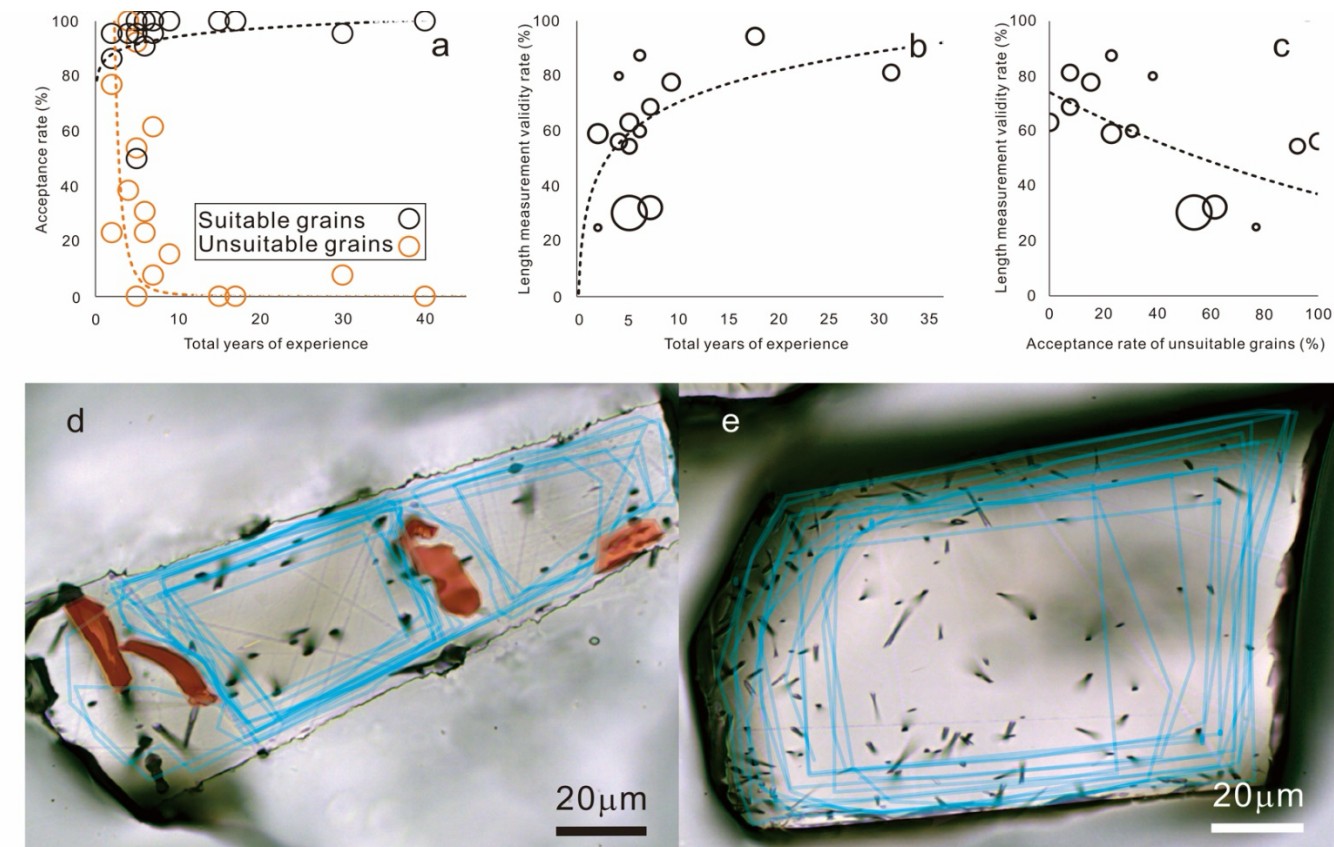

Figure 4: (a) Correlation of years of fission-track experience against acceptance percentage of suitable and unsuitable grains for density measurement and (b) confined track length measurement validity percentage. (c) The confined track length measurement validity percentage against the acceptance percentage of unsuitable grains. The size of the circles in b and c reflect the number of confined track length measurements. (d) Participant sections of the region of interest are juxtaposed in a defect/inclusion-rich grain and (e) a zoned grain.

Participant 5 yielded the lowest acceptance percentages of 50% for suitable grains and 0% for unsuitable grains. Participant 5 has been working exclusively on high-quality samples from one region for their entire fission-track experience (5 years), leading to selecting only the best-looking grains. This type of bias has been termed the 'mere-exposure effect' or 'familiarity principle', the tendency to develop preferences for things because they are familiar (Tversky and Kahneman, 1974). An analyst with narrower grain quality experience may miss available thermal history information by omitting objectively suitable, but

less familiar, grains.

### 3.6 Region of Interest (ROI) Selection

Track density measurements on defect/inclusion-free grains with homogeneous track distributions may not be greatly affected by different region of interest (ROI) selections, but the number and size of defects within a given ROI can cause an underestimation of density by obscuring tracks (Fig 4d). Similarly, the selection of high and low track density areas within a

zoned grain can yield widely varying density determinations (Fig 4e). A single-spot or even dual-spot laser ablation approach on such grains may result in a significant dispersion of dates depending on the analyst's ROI selection. Several participants placed ROIs in too-close proximity to the mineral border ($<\sim10$ μm, Fig 4 d,e), where track registration is below the required $4\pi$ geometry (Fleischer et al., 1975).

### 3.7 Light Source Utilization and a Case Correction on a Single Grain

To demonstrate some effects of the light source and ROI specification on density measurement, at the request of one participant, we reanalyzed their track density for Grain 07, which was significantly lower than most of the group. Figure 5 shows the comparison on a track-by-track basis. Perhaps importantly, this analyst revealed a preference for counting tracks in transmitted light only. However, counting tracks solely in transmitted-light images can cause an underestimation of the track density (Aslanian et al., 2022; Tamer and Ketcham 2023). Figure 5 a and b show the transmitted and reflected light image for

this particular grain, with the participant analysis in blue and re-analysis using images from both light sources in red with excluded counts in yellow. This reanalysis suggested that some genuine tracks had been overlooked, especially in the region on the right that is shaded in transmitted light. After adjustment of the ROI, the addition of the overlooked tracks in reflected light, and the exclusion of a track showing a defect-like pattern, the track density increased by ~35%.

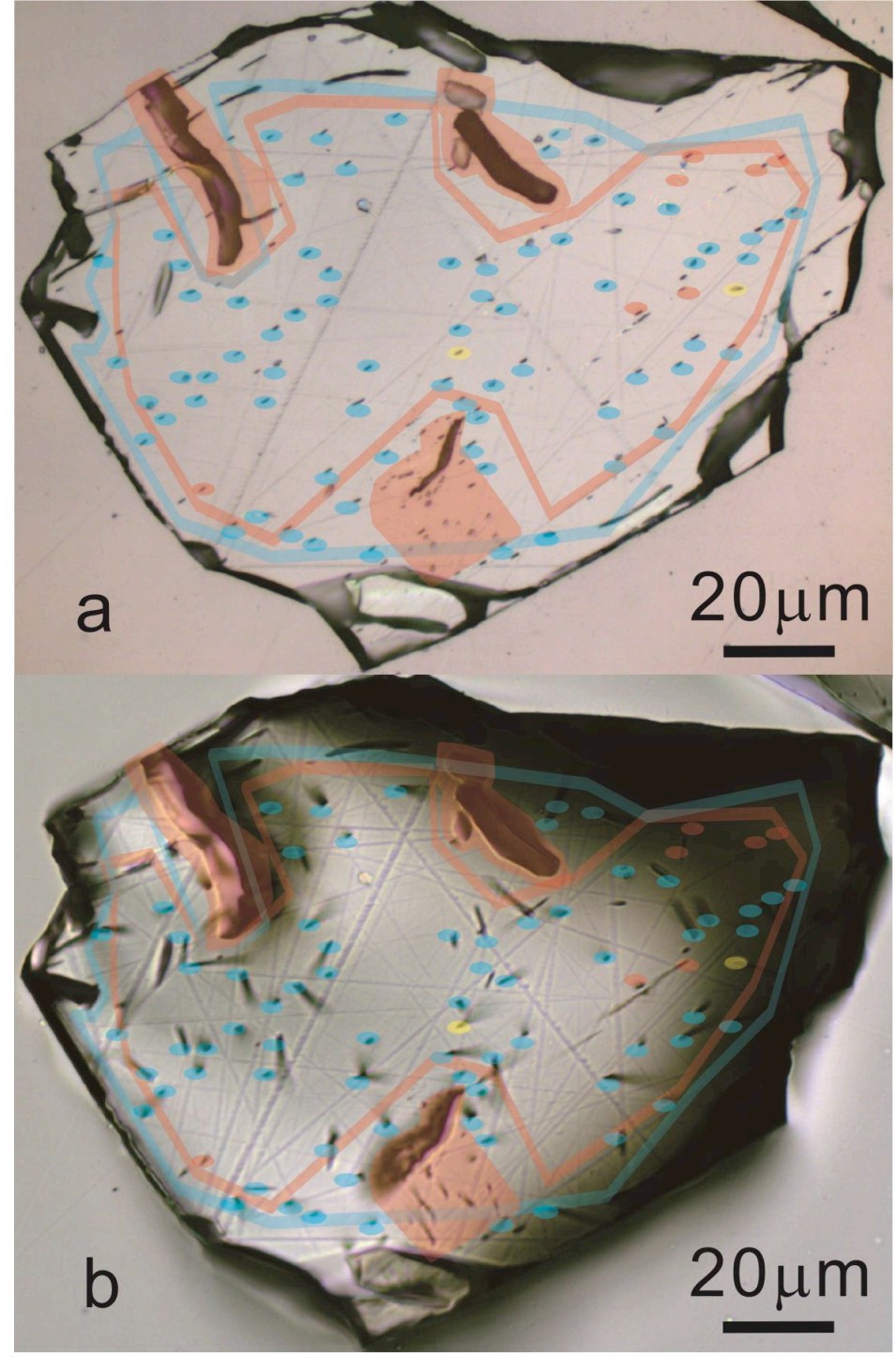

**Figure 5: Transmitted and reflected light images of Grain 7 (a, b) with the analysis of one participant (blue) and a reviewer's proposed correction for inclusion (red) and exclusion of tracks (yellow).**

### 3.8. Highlights of participant comments on the data set

Participants 1 and 5 found the images taken with no white balance to be not ideal for counting, and participants 5 and 7 mentioned that the grains etched with the 5.0 M HNO$_3$ 20s 20°C (Gleadow et al. 1986; Green et al., 1986) protocol appear to be under-etched.

### 4 Implications, Suggestions and Conclusions

Whether an analyst selects unsuitable grains and/or conducts invalid confined track length measurements will depend in part on their years of experience, and training, but may also be affected by the difficulty in finding sufficient grains to meet analytical goals. When the realities of a non-ideal sample conflict with an imposed requirement for how much data are required for study objectives, the result can end up being improper thermal history information. Comprehensive laboratory training and calibrations are essential for fission-track analysts to avoid these problems. Results of graticule and confined track length calibrations and the identity of the analyst should be stated in publications.

ROI selection may cause varying track density determinations, especially in zoned and defect-rich grains. While a single-spot laser ablation analysis is a time-efficient approach, its application on such grains may result in varying U determinations unless the laser spot covers a high proportion of the counted area (e.g., Cogné et al. 2020). Laser uranium mapping (Ansberque et al., 2021) or EDM (Gleadow and Lovering 1977) approaches require more work, but in ideal cases may better represent the selected region of interest. However, precise matching of spontaneous and induced track areas in the EDM can also be difficult in some cases. These approaches may also be more effective in identifying zoned grains when the spontaneous track density is low.

Zeta calibration ($\zeta$) using a set of age standards is intended to normalize uncertainties in some parameters in the age equation, such as thermal neutron fluence ($\phi$) and the spontaneous fission decay constant ($\lambda f$), and to account for varying counting efficiencies of different analysts (Hurford and Green 1983). This method assumes, however, that a calibration derived from measurements on near-ideal standard samples with minimal inclusions and defects and limited annealing (e.g, Durango, Fish Canyon Tuff apatites) also reflects analyst judgements in unknown samples, which may yield any quality of grains. Zoning, high levels of inclusions and defects, variable ROI selection, and even a more complex track length distribution may cause significant divergences unaccounted for by zeta calibration, particularly for a less-experienced analyst. These have been studied partially (Vermeesch 2017; Cogné and Gallagher 2021) but further work is needed on this matter. The analyst should remember that not all samples yield usable data.

Among the available confined tracks, analysts can select tracks with different effective etch times based on their individual perceptions and criteria. Some participants highlighted that the grains etched with 5.0 M $HNO_3$ at 20°C for 20s (e.g. Gleadow et al. 1986) appear under-etched, which agrees with a previous inter-analyst comparison experiment (Tamer et al., 2019). A proposed two-step etching protocol (5.5 $HNO_3$ 21°C 20+10s) allows analysts to select any suitable track but ensures that the final confined track length data set does not contain under-etched tracks (Tamer and Ketcham 2023). However, a protocol for modelling such data with existing annealing equations has not yet been put forward.

The application of AI and machine learning methods have become popular topics in various research fields in earth sciences including fission-track counting and confined track length measurements. The quality of any automated analyses will be defined by not only sophisticated algorithms but also the fission-track analysis experience of the initial "teacher" of the AI, including hardware preferences during image acquisition and the resulting image quality. Determining which grains and tracks are suitable for measurement represents an additional challenge for AI method development; training cannot be based on good images alone but must also include features that should be avoided based on sometimes subtle indications.

The accessibility of the "ingredients" of thermal history modeling is limited to data summary tables and sometimes raw data as supplementary files in research articles. Although fission-track data have generally fared well in inter-laboratory age comparisons in recent years, these have tended to utilize relatively straightforward samples. This study illustrates some of the potential hazards of fission-track analysis of more challenging materials, but also presents pathways toward improving data reliability. In particular, the opportunity is coming into view for the fission-track community to share data on a new level, allowing analysts to see, learn from, and discuss each others' image data.

Recent developments in data repositories and metadata reporting are healthy signs of an emerging open science culture and up-to-date reporting in low-T thermochronology. However, these are currently limited to collecting and presenting the data in their corresponding geo-locations (Boone et al., 2022; Boone et al., 2023a) and data reporting formats and table contents (Kohn et al., 2024). Given the continuing relevance of fission-track data, we recommend building toward a global infrastructure and culture enabling and encouraging data transparency and sharing through the formation of online digital image repositories (such as geochron@home (Vermeesch et al., 2023)), which can accommodate fission-track image data. Furthermore, proper analyst training and reconsideration of laboratory routines for image acquisition are needed. It has been over 60 years since fission-track dating method was first established (Price and Walker, 1962), and no clear guidelines have been formulated on "musts" and "cans" in fission-track practice. While existing fission-track laboratories develop their own preferences and routines, new laboratories often represent branching points, which can be a source of necessary and beneficial innovation but also undocumented and undesirable divergence. A global community repository housing guidelines for best-practice fission track analyses and fission track training modules are needed, as well as reference libraries of interpreted image sets. Adoption of an open science culture will ultimately benefit every fission-track laboratory and increase data quality.

*Code and data availability.* The image data used for this study, and the analysis results, have been uploaded to geochron@home (Vermeesch 2024). The analyst numbers in Table S1 are replaced with random letters in geochron@home to preserve anonymity.

*Author contributions.* Conceptualisation: MTT, LC, RAK, AJWG. Data curation: MTT, LC. Formal analysis: MTT, LC. Software: LC, AJWG. Investigation: MTT, LC. Methodology: MTT, LC, RAK, AJWG. Resources - MTT, LC, RAK, AJWG. Writing – original draft preparation: MTT, LC, RAK, AJWG. Writing – review& editing: MTT, LC, RAK, AJWG.

*Competing interests.* The authors have no competing interests to declare.

*Acknowledgements.* We appreciate the contributions of all of the participants to this effort. This research was supported by the Geology Foundation of the Jackson School of Geosciences and Fundamental Research Funds in the Institute of Geology, China Earthquake Administration (IGCEA2229), National Science Foundation of China (4231101318) and Project 3.51 of the AuScope Program within the Australian National Collaborative Research Infrastructure Strategy. We thank Raymond Donelick and Edward Sobel for the reviews and Shigeru Sueoka for editorial handling.

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
