# Peer review of "The Need for Fission Track Data Transparency and Sharing"

_Geochronology, 2024_

## Referee Comment (RC1)

**Review by Ray Donelick of Tamer et al. "The need for fission track data transparency"**

**General Comments**
- The data presented here are very important and need to be published. My strong preference is to "Accept with Major Revisions" this submission. Lacking major revisions such as suggested here, I urge this paper be Rejected by the Editor, re-written by the Authors, and re-submitted for review.
- Major revision required: The document and its arguments are poorly organized and the English wording and grammar needs to be improved significantly.
- Suggestion regarding re-organization:

  Lines 11-12: "We report a new image-based inter-analyst study to investigate fission-track grain selection and analysis by 13 participants from an image data set that included grains of variable quality."

  From the perspective of the inter-analyst study, organize the data, results, and recommendations around these 6+ "essential ingredients" or essential steps:
  (1) Select suitable apatite grain for age measurement
  (2) Select region of interest for fission track counting
  (3) Count fission tracks intersecting ROI surface
  (4) Measure confined fission track lengths
  (5) Measure Dpar, Cl, or other kinetic parameter
  (6) Measure uranium concentration
  a. EDM – Count induced fission tracks corresponding to ROI surface
  b. LAICPMS – Measure representative area of ROI surface
  +
  (7) Collect, archive, and share digital images to measure and permit re-measurement of fission track data

- I have not reviewed figures, tables, and their captions, pending major revision.
- I have not reviewed supplementary materials, pending major revision.
- I am willing to review a revised manuscript.

**Title**
- A new title is suggested: The Need for Apatite Fission Track Data Transparency and Sharing

**Abstract**
- Emphasize this study concerns apatite fission track data.
- Define data transparency and how you propose to implement it.
- Define data sharing and how you propose to implement it.
- Include list of 6+ "essential ingredients" or essential steps here and argue around them

Lines 16-17: "there is a danger of "squeezing the rock" weakening selection criteria." "squeezing the rock" presumably comes from "squeezing blood from a rock" (a saying recognized in USA at least). Please drop this statement. What you mean by "squeezing the rock" is actually more like "filling a line in a table".

Line 18: This statement "Juxtaposing selected regions of interest (ROIs) on the same grains indicates that zoned grains and grains with inclusions and defects yield varying track density estimates, indicating that ROI placement can be an influential factor." is just one of many statements that can be made here. Either list all of them or drop this one.

**1 Introduction**
- Emphasize this study concerns apatite fission track data.

Line 35: First mention of "apatite", yet the data presented here are from only apatite, and several "essential ingredients" are dominated by apatite studies.

Lines 28-30: I suggest this list of 6+ "essential ingredients"
(1) Select suitable apatite grain for age measurement
(2) Select region of interest for fission track counting
(3) Count fission tracks intersecting ROI surface
(4) Measure confined fission track lengths
(5) Measure Dpar, Cl, or other kinetic parameter
(6) Measure uranium concentration
a. EDM – Count induced fission tracks corresponding to ROI surface
b. LAICPMS – Measure representative area of ROI surface
+
(7) Collect, archive, and share digital images to measure and permit re-measurement of fission track data

Lines 29-30: "(4) average etch pit diameter (Dpar) measurements per grain," re-word to "(4) mean etch figure diameter parallel to c-axis (Dpar; Donelick, 1993; Burtner et al., 1994; Donelick et al., 1999) for each apatite grain," I hate being the guy pushing his own papers, but these papers are appropriate, especially given the >10 years Dr. Gleadow denied the value of this parameter (and its sister Dper).

Donelick, R.A., Ketcham, R.A., and Carlson, W.D., 1999, Variability of apatite fission track annealing kinetics II: Crystallographic orientation effects. American Mineralogist, v. 84, pp. 1224-1234.

Burtner, R.L., Nigrini, A., and Donelick, R.A.,1994, Thermochronology of Lower Cretaceous source rocks in the Idaho-Wyoming thrust belt. American Association of Petroleum Geologists Bulletin, vol. 78, no. 10, pp. 1613-1636.

Donelick, R.A., 1993, A method of fission track analysis utilizing bulk chemical etching of apatite. U.S. Patent Number 5,267,274.

Lines 31-35: "While laser ablation mass spectrometry has become an alternative (Hasebe et al., 2014) to the widely used external detector method (EDM) (Gleadow and Lovering 1977) for uranium content determination, the first four inputs are still largely analyst-driven, although recent developments in image analysis and AI have contributed significant advances in auto-counting and auto-measurement (Gleadow et al., 2009, 2019; Nachtergaele and De Grave 2021; Li et al., 2022; Ren et al., 2023; Boone et al., 2023)." Break this sentence into 2 or more sentences to make these points.

Line 36: "significant variation in measurements for the same samples and even standards" More information is needed here. The abstract in Line 1 leads off introducing your new "inter-analyst study" so you need to compare new to old. Help the reader better understand which of the "6+ essential

ingredients" might be the source of "significant variation" here or there in previous and the current work.

Line 43: "Grains where oily fluids have penetrated" It is not only "oily fluids", but also aqueous fluids. My experiences is that a paper towel alone cannot be guaranteed to remove distilled water in tracks from washing after etching.

Lines 46-47: "cause overestimation of ages." Or underestimation of the presence of defects here and there (say in Durango) may cause the analyst to lean toward "defect" for questionable features.

Lines 52-53: "Whereas the area counted for fission-track density determinations has typically been defined by boxes in an eyepiece reticule,…" Replace "typically" with "historically". "…modern image-based systems allow the user to draw an arbitrarily shaped region of interest." Replace "modern" with "recent". What is a reticule? The original sentence is condescending.

Line 56: "geometry (Donelick et al., 2005)" A better reference would be Fleischer et al. (1975).

Fleischer, R.L., Price, P.B., Walker, R.M., 1975, Nuclear Tracks in Solids: Principles and Techniques. University of California Press, Berkeley, 605 p.

Lines 64-65: "The suggested number of grains for age measurements for igneous-type samples is typically ~20, or more if there is any indication of kinetic variation (Donelick et al., 2005)" Donelick et al. (2005) did not suggest 20 grains ages, but merely stated that it was common practice. The source of 20 grain ages is Dr. Gleadow, with the choice of 20 grain ages being more concerned with making money (minimal work for Geotrack in the 80s) and less concerned with science (getting a pooled age of desired quality and precision).

Line 66: "squeeze the rock" Drop this saying that may make little sense to some people. Instead, focus on the poor data resulting from poor decisions.

**2 Materials and Methods**
Line 77: "41 grain and 3 graticule images" What do you mean by grain? What do you mean by graticule images? The word apatite does not appear in this paragraph!

Line 82: "grains from UM were etched with 5M HNO3 at 20°C for 20s (Gleadow et al. 1986)." I am pretty sure Dr. Gleadow has been under-etching c-axis-parallel fission tracks in F-rich apatites since the late 1970s. Please put the correct reference here.

Line 88: "and in relevant email lists". I was not included in your list. Who decides who gets invited?

Lines 88-100: Drop the sales pitch and tell us what this software does, and how the "experiment was made possible" with this software. I can do this study and much more with my own software, so I don't need Dr. Gleadow to enable me and my research. This reminds of the Iolite bait-and-switch.

Lines 105-106: "Rather they are simply used as reference values that are probably typical of reasonably experienced analysts." This paragraph is difficult as you do not want to tell everyone that your judgements are correct and those that deviate from yours are incorrect. Give the reader evidence here, right now, that you are qualified to make this decision. Show a zeta calibration with 100 grains of

Durango or something like that! Provide those images too. We should be requiring this routinely, you basically argue this point, and you do not provide the evidence.

Lines 113-114: "was considered unsuitable" By whom? I assume the answer is "L. Chung and… M. Tamer" from Lines 103-104.

Line 119: "borderline-quality grains" Borderline is not defined here. Suitable is not defined here. This whole paragraph needs to be flipped, tell us what makes a grain suitable, what is compromised for borderline, and then finish with what makes a grain absolutely unsuitable.

Line 154: "Graticule images" Really? Images of the graticule in the eyepiece? Perhaps you mean images of a NIST-traceable length calibration grid on a microscope slide?

Line 155: "only five participants reported measuring them." Well, did they get similar results to the default graticule calibration? More info please.

Lines 161-162: "varying region of interest selection, light source preference, and track counting routines" You discuss varying region of interest selection. What do you mean by light source preference and do you have data to back this up? Same question for track counting routines, after telling us what you mean by track counting routines.

Lines 180-181: "This high rate of acknowledgment by the participants supports the soundness of the criteria utilized by the reviewers." Maybe. Who cares. What matters is that there is room to educate each other in this field and to, perhaps, lower variation among labs by abiding by the principles of data transparency and data sharing.

Lines 185-186: "Some of the participants used FastTracks' automatic tools for c-axis orientation and dpar length measurements." This is important information out of nowhere. You need to separate out the effects of these measurements from those who did not use these tools. Also, you should show how well FastTracks reproduced measurements from analyst to analyst that used these capabilities. The ultimate goal is to lower variance among analysts. Did FastTracks succeed or fail here?

Lines 196-197: "Participants 1 and 10 and Participants 8 and 9 are from the same two laboratories and show similarities in their respective track density results." Give numbers here and elsewhere in this discussion that back up your statements. Don't make me search for this information in the tables and figures.

Lines 216-228: The discussion does not give any numbers telling the reader what is meant by "relatable and consistent", "skewed… to lower values", "varying number of measurements", "show similarities", "divergent results". Please use the results to make your case. These generalized statements teach me nothing.

Line 236: "Donelick et al., 2005" A better reference is Fleischer et al., 1975.

Lines 256-257: "However, counting tracks solely in transmitted-light images can cause an underestimation of the track density (Aslanian et al., 2022; Tamer and Ketcham 2023)." The word "can" does not mean "does". And I don't' need Dr. Ketcham telling me how to count. What matters more is that analysts apply the same methods/criteria/data types to unknowns as they do standards. The

argument can he made that data from this whole paper need to be divided by data from the next-inter-analyst study of appropriate age and length calibration standards.

Line 268: "grains etched with the 5.0 M HNO3 20s 20☐C (Gleadow et al. 1986) protocol appear to be under-etched" Because they are. They have been since 1977 or so. They continue to be.

Lines 27-271: "Analysts may consider unsuitable grains and/or conduct invalid confined track length measurements depending on their years of experience, training, and the difficulty in finding sufficient grains to meet analytical goals." This sentence needs to be re-written so that the several points being made are clear to the reader.

Line 271: ""Squeezing the rock"," Drop this saying.

Line 274: "Results of graticule and confined track length calibrations and the identity of the analyst should be stated in publications." In this paper, you offer graticule calibrations somewhere. You do not offer any confined length calibration data. You don't even mention, much less offer, any age calibration data such a zeta calibration standard. I would like to see data here divided by the appropriate calibration data.

Line 280-281: "precise matching of spontaneous and induced track areas in the EDM can also be difficult in some cases." I would love to sort through the decades of mica detectors affixed to under-etched AFT grain mounts at UMelbourne and elsewhere and reveal the staggering percentage of EDM images that are poor due to poor contact – but counted anyway to produce a line in a data table.

Lines 306-307: "Although fission-track data have generally fared well in inter-laboratory age comparisons in recent years" My assessment is just the opposite. The variance among laboratories is increasing, not decreasing, since the 1980s. This is almost certainly due to inconsistent – perhaps even poor at times – training of analysts, at the start and as the years go by.  This is made easier by flashy hardware and software products that give the appearance of expertise but do not substitute for it.

Line 316: "encouraging data transparency" Re-write to "encouraging data transparency and sharing".

---

## Author Response (AR1)

A point-by-point response to the reviewer comments

Reviewer 1

General Comments

- The data presented here are very important and need to be published. My strong preference is to "Accept with Major Revisions" this submission. Lacking major revisions such as suggested here, I urge this paper be Rejected by the Editor, re-written by the Authors, and re-submitted for review.

- Major revision required: The document and its arguments are poorly organized and the English wording and grammar needs to be improved significantly.

- Suggestion regarding re-organization:

Lines 11-12: "We report a new image-based inter-analyst study to investigate fission-track grain selection and analysis by 13 participants from an image data set that included grains of variable quality."

From the perspective of the inter-analyst study, organize the data, results, and recommendations around these 6+ "essential ingredients" or essential steps:

(1) Select suitable apatite grain for age measurement

(2) Select region of interest for fission track counting

(3) Count fission tracks intersecting ROI surface

(4) Measure confined fission track lengths

(5) Measure Dpar, Cl, or other kinetic parameter

(6) Measure uranium concentration

a.EDM – Count induced fission tracks corresponding to ROI surface

b.LAICPMS – Measure representative area of ROI surface

+

(7) Collect, archive, and share digital images to measure and permit re-measurement of fission track data

- I have not reviewed figures, tables, and their captions, pending major revision.

- I have not reviewed supplementary materials, pending major revision.

- I am willing to review a revised manuscript.

*Our Reply: We thank the reviewer for the comments and suggestions. We implemented almost all of the reviewer suggestions and rejected some others, including re-organization. The reviewer documents seven essential ingredients and asks to reorganize the data, results and recommendations around these 6+ essential ingredients. From these 7 points, we focused on #1, #2, #3, #4 and #5 in our investigations and suggested the 7th point as part of our conclusions. We did review a single grain of an analyst in Figure 5 that covers #3 listed in the reviewers list. The results of Dpar measurements (#5) are also documented in the supplementary files. Uranium concentration (6#) was not in the scope of this study. From our perspective the current state of construction of this study clearly covers and explains all these points already. We edited the introduction according to reviewers list.*

*We provide the reasons for those suggestions we rejected to adopt for we hope to promote healthier scientific communication with the reviewer and constructive criticism for the benefit of the readers and our study here. In the abstract, we briefly document the study, its major results, and conclusions. Documenting and arguing about these 7 "essential ingredients" in the abstract would make it unnecessarily longer.*

Title

• A new title is suggested: The Need for Apatite Fission Track Data Transparency and Sharing

*Our Reply: We changed the title accordingly.*

Abstract

- Emphasize this study concerns apatite fission track data.

*Our Reply: Edited.*

- Define data transparency and how you propose to implement it.

*Our Reply: Edited.*

- Define data sharing and how you propose to implement it.

*Our Reply: Edited.*

- Include list of 6+ "essential ingredients" or essential steps here and argue around them

*Our Reply: The major motivation of this study is to compare grain selection, ROI placement, length measurement validity and overall comparison of Dpar measurements. Listing and arguing on these 6+ essential ingredients do not belong to the abstract in our opinion.*

Lines 16-17: "there is a danger of "squeezing the rock" weakening selection criteria." "squeezing the rock" presumably comes from "squeezing blood from a rock" (a saying recognized in USA at least). Please drop this statement. What you mean by "squeezing the rock" is more like "filling a line in a table".

*Our Reply: Edited.*

Line 18: This statement "Juxtaposing selected regions of interest (ROIs) on the same grains indicates that zoned grains and grains with inclusions and defects yield varying track density estimates, indicating that ROI placement can be an influential factor." is just one of many statements that can be made here. Either list all of them or drop this one.

*Our Reply: All the factors are listed in the introduction. In this study, we examined the ROI for the first time, so, this statement should be in the abstract.*

1 Introduction

• Emphasize this study concerns apatite fission track data.

*Our Reply: Edited.*

Line 35: First mention of "apatite", yet the data presented here are from only apatite, and several "essential ingredients" are dominated by apatite studies.

*Our Reply: We added apatite in the first sentence of introduction.*

Lines 28-30: I suggest this list of 6+ "essential ingredients"

(1) Select suitable apatite grain for age measurement

(2) Select region of interest for fission track counting

(3) Count fission tracks intersecting ROI surface

(4) Measure confined fission track lengths

(5) Measure Dpar, Cl, or other kinetic parameter

(6) Measure uranium concentration

a.EDM – Count induced fission tracks corresponding to ROI surface

b.LAICPMS – Measure representative area of ROI surface

+

(7) Collect, archive, and share digital images to measure and permit re-measurement of fission track data

*The first ingredient listed by the reviewer is the pre-condition of fission track studies, and the last one is our conclusion. We like to stick to the current list as it is but we edited the text according to reviewer comments.*

Lines 29-30: "(4) average etch pit diameter (Dpar) measurements per grain," re-word to "(4) mean etch figure diameter parallel to c-axis (Dpar; Donelick, 1993; Burtner et al., 1994; Donelick et al., 1999) for each apatite grain," I hate being the guy pushing his own papers, but these papers are appropriate, especially given the >10 years Dr. Gleadow denied the value of this parameter (and its sister Dper).

Donelick, R.A., Ketcham, R.A., and Carlson, W.D., 1999, Variability of apatite fission track annealing kinetics II: Crystallographic orientation effects. American Mineralogist, v. 84, pp. 1224-1234.

Burtner, R.L., Nigrini, A., and Donelick, R.A.,1994, Thermochronology of Lower Cretaceous source rocks in the Idaho-Wyoming thrust belt. American Association of Petroleum Geologists Bulletin, vol. 78, no. 10, pp. 1613-1636.

Donelick, R.A., 1993, A method of fission track analysis utilizing bulk chemical etching of apatite. U.S. Patent Number 5,267,274.

*Our Reply: Edited..*

Lines 31-35: "While laser ablation mass spectrometry has become an alternative (Hasebe et al., 2014) to the widely used external detector method (EDM) (Gleadow and Lovering 1977) for uranium content determination, the first four inputs are still largely analyst-driven, although recent developments in image analysis and AI have contributed significant advances in auto-counting and auto-measurement (Gleadow et al., 2009, 2019; Nachtergaele and De Grave 2021; Li et al., 2022; Ren et al., 2023; Boone et al., 2023)." Break this sentence into 2 or more sentences to make these points.

*Our Reply: Edited.*

Line 36: "significant variation in measurements for the same samples and even standards" More information is needed here. The abstract in Line 1 leads off introducing your new "inter-analyst study" so you need to compare new to old. Help the reader better understand which of the "6+ essential ingredients" might be the source of "significant variation" here or there in previous and the current work.

*We cannot catalogue and compare all of these studies, but we have noted what they lack in relation to the current one, and relate these to the listed ingredients.*

Line 43: "Grains where oily fluids have penetrated" It is not only "oily fluids", but also aqueous fluids. My experiences is that a paper towel alone cannot be guaranteed to remove distilled water in tracks from washing after etching.

*Our Reply: Edited.*

Lines 46-47: "cause overestimation of ages." Or underestimation of the presence of defects here and there (say in Durango) may cause the analyst to lean toward "defect" for questionable features.

*Our Reply: Excellent point that we missed. Edited. Thank you!*

Lines 52-53: "Whereas the area counted for fission-track density determinations has typically been defined by boxes in an eyepiece reticule,..." Replace "typically" with "historically". "...modern image- based systems allow the user to draw an arbitrarily shaped region of interest." Replace "modern" with "recent". What is a reticule? The original sentence is condescending.

*Our Reply: Edited.*

Line 56: "geometry (Donelick et al., 2005)" A better reference would be Fleischer et al. (1975).

Fleischer, R.L., Price, P.B., Walker, R.M., 1975, Nuclear Tracks in Solids: Principles and Techniques. University of California Press, Berkeley, 605 p.

*Our Reply: Edited.*

Lines 64-65: "The suggested number of grains for age measurements for igneous-type samples is typically ~20, or more if there is any indication of kinetic variation (Donelick et al., 2005)" Donelick et al. (2005) did not suggest 20 grains ages, but merely stated that it was common practice. The source of 20 grain ages is Dr. Gleadow, with the choice of 20 grain ages being more concerned with making money (minimal work for Geotrack in the 80s) and less concerned with science (getting a pooled age of desired quality and precision).

*Our Reply: We changed the text to no longer imply that 20 was a suggestion by Donelick (2005). We have not been able to find the source of the number 20, but note that Wagner and van den Haute (1992) state "often a number (n) of 10 grains or more is analyzed in order to have a good statistical sample" (section 3.8.2, p. 85). We also note that the attribution in this comment is inappropriate as well as unsupported.*

Line 66: "squeeze the rock" Drop this saying that may make little sense to some people. Instead, focus on the poor data resulting from poor decisions.

*Our Reply: Edited.*

2 Materials and Methods

Line 77: "41 grain and 3 graticule images" What do you mean by grain? What do you mean by graticule images? The word apatite does not appear in this paragraph!

*Our Reply: Edited.*

Line 82: "grains from UM were etched with 5M HNO3 at 20°C for 20s (Gleadow et al. 1986)." I am pretty sure Dr. Gleadow has been under-etching c-axis-parallel fission tracks in F-rich apatites since the late 1970s. Please put the correct reference here.

*Our Reply: This comment is, again, unprofessional and off-point. Gleadow and Lovering (1978) indeed used this same protocol, but that study only measured track densities, not lengths. We cite Gleadow et al. (1986) as the one that proposed this protocol for confined length measurements, and added a citation to Green et al (1986), which used step etching to support its appropriateness for generating the first confined-length-annealing data set that served as the foundation of the first thermal-history modeling from apatite fission tracks.*

Line 88: "and in relevant email lists". I was not included in your list. Who decides who gets invited?

*Our Reply: The announcement of this study was made to public during 17th International Conference of Thermochronology, Santa Fe (Thermo2023), followed by email invitations to the Thermo2023 and geo-tectonics jiscmail email lists. We apologize to anyone we could not reach.*

Lines 88-100: Drop the sales pitch and tell us what this software does, and how the "experiment was made possible" with this software. I can do this study and much more with my own software, so I don't need Dr. Gleadow to enable me and my research. This reminds of the Iolite bait-and-switch.

*Our Reply: We believe in freedom other than "my way or no way" approach. No participant was preconditioned to use any specific software tool. The second sentence of the previous paragraph in the manuscript and the last part of the study announcement (https://doi.org/10.1002/essoar.10507907.2) clearly states that anyone could participate in this study using any software tool. For those who do not have FastTracks but would like to*

*participate using it, we provided a limited license for participation. We hope to see reviewer's participation with his own software in the future inter-analyst studies. If the reviewer would provide a limited software tool just for participation, we would like to test it in the future studies. In the same paragraph towards to the end of section 2.2 we explain that the .xml files can be used reload the analyses.*

Lines 105-106: "Rather they are simply used as reference values that are probably typical of reasonably experienced analysts." This paragraph is difficult as you do not want to tell everyone that your judgements are correct and those that deviate from yours are incorrect. Give the reader evidence here, right now, that you are qualified to make this decision. Show a zeta calibration with 100 grains of Durango or something like that! Provide those images too. We should be requiring this routinely, you basically argue this point, and you do not provide the evidence.

*Our Reply: Reviewers of this study rejected each other's track length measurements by 16% in the previous inter-analyst study (Tamer et al., 2019), which is reduced to ~1% in this study. We believe that the reviewers of this study conducted measurements, failed in some, learned lessons and improved with time. We do not claim that those that deviate from our measurements are incorrect per se, however, if the deviation is significant there must be some reasons behind it. Having a different opinion on suitability of a few grains and maybe some tracks is an expected outcome. In fact, we think that if all the analysts of ~50 fission track labs would participate in a larger scale global study, there will be no 100% consensus on all the aspects of fission track analysis. In this study, we rather point out the deviations where we see them the most. To tackle these, we are preparing fission track analysis guidelines and teaching modules as future studies for the fission track community.*

Lines 113-114: "was considered unsuitable" By whom? I assume the answer is "L. Chung and... M. Tamer" from Lines 103-104.

*Our Reply: Edited.*

Line 119: "borderline-quality grains" Borderline is not defined here. Suitable is not defined here. This whole paragraph needs to be flipped, tell us what makes a grain suitable, what is compromised for borderline, and then finish with what makes a grain absolutely unsuitable.

*Our Reply: By defining the unsuitable, we define the suitable. We like to stick to this construction and added additional explanation for borderline-quality grains.*

Line 154: "Graticule images" Really? Images of the graticule in the eyepiece? Perhaps you mean images of a NIST-traceable length calibration grid on a microscope slide?

*Our Reply: In section 2.1 we defined graticule as length calibration grid on a microscope slide according to one of the previous comments of the reviewer.*

Line 155: "only five participants reported measuring them." Well, did they get similar results to the default graticule calibration? More info please.

*Our Reply: Edited.*

Lines 161-162: "varying region of interest selection, light source preference, and track counting routines" You discuss varying region of interest selection. What do you mean by light source preference and do you have data to back this up? Same question for track counting routines, after telling us what you mean by track counting routines.

*Our Reply: The analyses on these grains show different ROI placements. During the follow-up some of the participants mentioned that they use transmitted light or reflected light only. Combination of these two pieces of information led us to mention track counting routines.*

Lines 180-181: "This high rate of acknowledgment by the participants supports the soundness of the criteria utilized by the reviewers." Maybe. Who cares. What matters is that there is room to educate each other in this field and to, perhaps, lower variation among labs by abiding by the principles of data transparency and data sharing.

*Our Reply: Thank you for the statement. This high rate of acknowledgment rather suggests that there is a need of guidelines and teaching modules.  Lowering the variation among the labs will be the ultimate goal in some of our future studies, where we hope to see reviewer's participation in the future inter-analyst studies.*

Lines 185-186: "Some of the participants used FastTracks' automatic tools for c-axis orientation and dpar length measurements." This is important information out of nowhere. You need to separate out the effects of these measurements from those who did not use these tools. Also, you should show how well FastTracks reproduced measurements from analyst to analyst that used these capabilities. The ultimate goal is to lower variance among analysts. Did FastTracks succeed or fail here?

*Our Reply:* We did not make this comparison because during the follow-ups we failed to ask the participants if they corrected auto-measurements. This was poor follow-up discussion planning on our part.

Lines 196-197: "Participants 1 and 10 and Participants 8 and 9 are from the same two laboratories and show similarities in their respective track density results." Give numbers here

and elsewhere in this discussion that back up your statements. Don't make me search for this information in the tables and figures.

*Our Reply: In Figure 3 we provide density histograms from all the participants, ranging from 0 to 2.0 $10^6$ track/$cm^2$). We suggest the reviewer to follow up with the figure, since providing a single mean or median density value may be misleading.*

Lines 216-228: The discussion does not give any numbers telling the reader what is meant by "relatable and consistent", "skewed... to lower values", "varying number of measurements", "show similarities", "divergent results". Please use the results to make your case. These generalized statements teach me nothing.

*Our Reply: We provided two additional tables and implemented some of the general/more important values in the text.*

Line 236: "Donelick et al., 2005" A better reference is Fleischer et al., 1975.

*Our Reply: Edited.*

Lines 256-257: "However, counting tracks solely in transmitted-light images can cause an underestimation of the track density (Aslanian et al., 2022; Tamer and Ketcham 2023)." The word "can" does not mean "does". And I don't need Dr. Ketcham telling me how to count. What matters more is that analysts apply the same methods/criteria/data types to unknowns as they do standards. The argument can he made that data from this whole paper need to be divided by data from the next-inter- analyst study of appropriate age and length calibration standards.

*Our Reply: We do not presume to be telling Dr. Donelick how to count, we are merely conveying the findings of these two, independent, studies.*

Line 268: "grains etched with the 5.0 M HNO3 20s 20 C (Gleadow et al. 1986) protocol appear to be under-etched" Because they are. They have been since 1977 or so. They continue to be.

*Our Reply: Again, the comment is inappropriate. Our view based on other work is that all etching protocols produce under-etched tracks, because confined track revelation is a continuous process, with new tracks being intersected by expanding etchant pathways at all times. The relevant questions are what proportion of visible tracks are sufficiently etched, and how consistent are analyst choices. But, that's not an argument for this paper.*

Lines 27-271: "Analysts may consider unsuitable grains and/or conduct invalid confined track length measurements depending on their years of experience, training, and the difficulty in

finding sufficient grains to meet analytical goals." This sentence needs to be re-written so that the several points being made are clear to the reader.

*Our Reply: We have edited this text.*

Line 271: ""Squeezing the rock"," Drop this saying.

*Our Reply: Edited.*

Line 274: "Results of graticule and confined track length calibrations and the identity of the analyst should be stated in publications." In this paper, you offer graticule calibrations somewhere. You do not offer any confined length calibration data. You don't even mention, much less offer, any age calibration data such a zeta calibration standard. I would like to see data here divided by the appropriate calibration data.

*Our Reply: This is another poor planning on our side, we should have asked about the confined track length standard calibration results from the participants at the beginning. Graticule, confined track length calibrations and the identity of the analyst are not requested by the journals and editors but we hope that with this study, journal editors will slowly require this info along with images and analyses. We agree with the comment however, zeta calibration was beyond the scope of this study. This will be surely part of the next inter-analyst study.*

Line 280-281: "precise matching of spontaneous and induced track areas in the EDM can also be difficult in some cases." I would love to sort through the decades of mica detectors affixed to under-etched AFT grain mounts at UMelbourne and elsewhere and reveal the staggering percentage of EDM images that are poor due to poor contact – but counted anyway to produce a line in a data table.

*Our Reply: This is another comment that is neither appropriate nor germane.*

Lines 306-307: "Although fission-track data have generally fared well in inter-laboratory age comparisons in recent years" My assessment is just the opposite. The variance among laboratories is increasing, not decreasing, since the 1980s. This is almost certainly due to inconsistent – perhaps even poor at times – training of analysts, at the start and as the years go by. This is made easier by flashy hardware and software products that give the appearance of expertise but do not substitute for it.

*Our Reply: We disagree. According to our own analysis the reproducibility observed in the data submitted for Ketcham et al. (2018) compares favorably with that among participants reported by Miller et al. (1985, 1990, 1993). We are unsure of which aspects of the data the reviewer is focusing on, or what statistics he is using, to arrive at his conclusion.*

Line 316: "encouraging data transparency" Re-write to "encouraging data transparency and sharing".

*Our Reply: Edited.*

Reviewer 2

Review by Ed Sobel of Tamer et al. "The Need for Fission Track Data Transparency"

This study presents an inter-operator study on the quality of fission track analyses; the influence of the operator's skill and judgement on the results are the main focus. After revisions (English usage and more precise descriptions), this will be a very useful study for the fission track community.

I have read Ray Donelick's comments and generally agree with his points. However, I think that the required revisions are primarily textural and therefore are minor rather than major/reject. I would modify his list of 6+ essential steps: 7 (collect photos) has to come before 6b (LA-ICP-MS). I agree with his wish to have more precise statements. Yes, the number of analysts is small, but quantitative statements would be more useful than phrases such as 'show similarities'. The criteria for suitable, unsuitable, or borderline needs to be defined - the text is presently quite short, so there is space to elaborate. Fig. 3 shows data histograms. A more quantitative way to compare histograms would be helpful.

*Our Reply: We thank the reviewer for the comments and suggestions. We implemented all the suggestions, including reorienting the figures to portrait format, inclusion of Table S1 to the main text and other textural edits.*

47  Track density can vary by up to 35% if the grain is oriented without the c-axis in the viewing plane

Rephrase: if the grain is not oriented with the c-axis in the viewing plane

*Our Reply: Edited.*

58-59  and suffer edge effects from sampling a 4pi region that is variable.

rephrase-perhaps: that hosts variable U concentration

*Our Reply: Edited.*

A follow-on conclusion would be that software packages should be modified to provide the option of automatically drawing an ROI that is 10 micron inside the edge of the grain.

*Our Reply: Next version of FastTracks (Ling et al. in prep) will include this feature.*

~80 Please insert a sentence here stating where the images used in this study can be viewed. Presently one has to reach line 326 to get this information.

*Our Reply: Edited.*

Table 1 - I don't understand the suitable grain selection rate. Analyst 1 selected 22 grains as suitable and has a 100% rate; analyst 8: 35 grains, 100%. I find the description of the selection criteria to be confusing.

*Our Reply: We implemented new tables and improved the text in section 3.5 for further clarifications.*

Rate is the wrong term. A number of grains were selected - this is a scalar unit. Rate implies a speed. This error occurs throughout the ms.

*Our Reply: Edited.*

Figs. 1, 3 - rearrange so that these are not in landscape format.

*Our Reply: Edited.*

The figures and tables are well drawn and appropriate.

*Our Reply: Thank you for the comment.*

Fig. S1 could be included in the main ms. Table S2 needs a title.

*Our Reply: Edited.*

154-5  Although calibration is an essential step before performing an analysis,

   This needs justification. How large were the differences in lengths between groups that did and did not perform calibrations? I note that my (old) microscope is quite stable. The change from one calibration to the next is too small to measurably affect lengths -  on the order of 0.02-0.03 microns for a 15 micron length and less for a shorter length, which is within measurement error.

*Our Reply: This is an important question, however, the experimental design of our study cannot provide a satisfactory answer. We cannot directly compare mean track lengths because different analysts measured different lengths and different number of lengths, inclusion or exclusion of*

*few tracks into a data set may drastically change average mean, stdev, min and max values. The only suitable way is to make a comparison on measurements of single track lengths. There is no confined track length measured by all the analysts in this study. In the figure below we show a case comparison of measurements on a single confined track (Figure a,b) by 11 analysts. For this particular track, the apparent lengths (2D) range from ~11.5 to 12.1 μm, where most of the measurements are within 1 sigma of the mean value (red circle) (Figure c). The difference in true lengths (3D) is slightly higher (Figure d) than 2D length due to different considerations on dip angle (Figure b,e). Based on one single track, there is no correlation between those who measured graticule and those who did not. However, we cannot draw a relationship on a single length. There are several other confined track lengths measured by lesser number of analysts that we can compare and not all of them are measured by the analysts who measured graticules. Few length measurements repeated by lesser number of analysts does not constitute a solid ground to answer this question properly.*

[Figure]

*Figure: Comparison of a single confined track length measured by 11 analysts. Confined track length (a), its cross section view (b), comparison of the appearent (2D) length (c), true (3D) length (d) and dip angle (e). Solid lines show 1 sigma intervals from the average value.*

185-6: Some of the participants used FastTracks' automatic tools for c-axis orientation and Dpar length measurements.

this sentence doesn't match the section header (3.3 Post-review follow-up and objectivity of the review). It belongs somewhere else.

*Our Reply: The sentence is moved to section 3.*

189 The density 190 distributions of suitable grains appear to be somewhat more relatable...

more relatable is the wrong phrase. similar?  In general, the writing in section 3.4 is rough.

*Our Reply: Edited.*

section 3.5 new text: "Accordingly, those who admit unsuitable grains at higher rates "

Admit is the wrong word - select?

*Our Reply: Edited.*

231 ...but inclusion of large defects in the ROI...

a large number? I don't think the size of the individual defects is as relevant as their abundance.

*Our Reply: Both number and size of the defects contribute the area that is unable to be analyzed within a given ROI. We edited the sentence.*

148    reasulting typo

*Our Reply: Edited.*

Does the paper address relevant scientific questions within the scope of GChron?

   Absolutely

*Our Reply: Thank you.*

Does the paper present novel concepts, ideas, tools, or data?

   The paper presents the novel study in which multiple users analyze the exact same apatite fission track crystals. This leads to observations and conclusions about the causes of variability and the reliability of measurements.

*Our Reply: Thank you.*

Are the scientific methods and assumptions valid and clearly outlined?

   Can be improved - the criteria for grain selection need to be better defined.

*Our Reply: Edited.*

Are the results sufficient to support the interpretations and conclusions?

It would be better is some qualified observations were quantified, as discussed above.

*Our Reply: Edited.*

Is the description of experiments and calculations sufficiently complete and precise to allow their reproduction by fellow scientists (traceability of results)?

yes

*Our Reply: Thank you.*

Do the authors give proper credit to related work and clearly indicate their own new/original contribution?

yes

*Our Reply: Thank you.*

Does the title clearly reflect the contents of the paper?

ok. Donelick's suggested title would be an improvement.

*Our Reply: Edited.*

Does the abstract provide a concise and complete summary?

ok. Needs to include the word apatite in the 1st sentence.

*Our Reply: Edited.*

Is the overall presentation well structured and clear?

yes

*Our Reply: Thank you.*

Is the language fluent and precise?

Some English polishing is needed. The language usage light source preference can be imprecise. An example: "light source preference", which actually seems to mean the decision to use reflected as well as transmitted light. The definition of this phrase needs to be clearly stated.

*Our Reply: Edited.*

Are mathematical formulae, symbols, abbreviations, and units correctly defined and used?

Yes - with the exception of the word 'rate', as discussed above.

*Our Reply: Edited.*

Should any parts of the paper (text, formulae, figures, tables) be clarified, reduced, combined, or eliminated?

on

*Our Reply: We assume the answer is no.*

Are the number and quality of references appropriate?

yes

*Our Reply: Thank you.*

Is the amount and quality of supplementary material appropriate?

yes

*Our Reply: Thank you.*

A point by point response to the handling editor

MS No.      : gchron-2024-26
Title         : The Need for Fission Track Data Transparency
Author(s)   : Murat T. Tamer et al.

**Comments to the Author:**

Thank you for submitting your manuscript to GChron. The two reviewers agreed in principle on the value of the paper, but differed somewhat in their assessment. Reviewer 2 suggested a major revision mainly because of the need to reorganize the manuscript, while reviewer 1 suggested that a minor revision is more appropriate since it is a compositional issue anyway. My opinions in this point and some other comments are as follows, and I believe that this manuscript is potentially suitable for publication in the GChron journal after minor to moderate revisions.

*Our response: We thank the handling editor, we adopted most of the suggestions and rejected a few of them, and we provide explanations for those we rejected.*

➢ I agree that the 6+ points mentioned by reviewer 2 are essential elements of the FT analysis, but I do not think that points not fully addressed in this manuscript need to be included in the structure of this manuscript. It would be sufficient to first state the 6+ essential ingredients of FT analysis in the introduction part (not in abstract) and then state which of them are mainly addressed in this manuscript (e.g., in L69-73). This will give the readers an overall picture of the FT analysis procedure and where this manuscript contributes or not.

*Our response: We reconstructed the introduction and added these points.*

➢ A common point made by the two reviewers is the recommendation to define the criteria of suitable/unsuitable/borderline grains and to use more quantitative expressions in describing the characteristics of the reported FT data. This is an important point in objectively evaluating the validity of the results/interpretations in this paper. The authors responded that they have added some important values to the main text and provided new tables to address this point. Although I cannot access the revised manuscript at this stage, I hope that this point has been carefully addressed.

*Our response: We extended the sentence regarding the borderline grains. By defining unsuitable grains in details we defined the suitable grains.*

➢ Reviewer 2 pointed out that etching apatite with 5M HNO3 at 20C for 20 seconds results in under-etching, but this seems to be beyond the scope of this manuscript. Since this etching recipe is widely used in FT labs around the world, this issue is too important to be discussed here. It is appropriate to provide further discussion in another paper with presenting sufficient data and evidence. The authors do not need to address the comments on this and similar issues.

*Our response: Under-etching of 5M HNO3 at 20C for 20s was an observation of a previous study (Tamer et al., 2019), which was echoed by the participants of this study as well. We added this statement as one of the highlights of participant comments. We prefer to keep this statement to reflect the voice of the participants.*

In addition to the referees' comments, I will make some minor comments and technical corrections. Please consider these as well.

L20-21 I think this point is not clearly stated in the main text. The only mention of the zeta calibration in Chapter 4 is in L284-291, where the selection bias caused by differences between near-ideal standards and actual unknowns is mainly focused on, rather than improvements of the zeta method related to differences in uranium measurements and etching protocols.

*Our response: Zeta is mentioned in the abstract because it is mentioned in the conclusions. In our study, we did not examine the zeta method. However, we see that there is a need for new approaches to the zeta method.*

L25 Balestrieri et al. 1999 should be 1991 (cf. L356-358)

*Our response: Edited.*

L31-32 Hasebe et al. 2014 should be 2004 (cf. L420-421)

*Our response: Edited.*

L37 2003a --> 2003

*Our response: Edited.*

L278 Cogne et al. 2020 is missing from the reference list.
*Our response: Edited.*

Table 1 It would be better to include the definition of the selection rates and validity rate in the caption. Without a clear denominator and numerator, it is difficult to understand how each number is viewed. Especially for "unsuitable grain selection rate", it is difficult to know whether a larger or smaller number is better.

*Our response: We added. After changing the "rates" into "percentages" and adding explanations to the table captions regarding how the percentages are calculated, we believe tables are clearer now.*

Figure 3 The graphs are arranged horizontally, but they would be easier to read if they were arranged vertically. For example, if the graphs are arranged in order from the left column, FT density for participants 1-8, FT density for participants 9-, FT length for participants 1-8, and FT length for participants 9-, the same types of graphs are arranged vertically, which makes it easier to see comparisons between participants.

*Our response: All the figures and tables are arranged vertically now.*

Reference        Green 1981 is not cited in the main body.

*Our response: Edited.*

Reference        There are two "Boone et al. 2023". Distinguish between the two by adding a and b.

*Our response: Edited.*

Whole text    Superscripts and subscripts do not seem to be reflected correctly in some terms, such as, $D_{par}$ (e.g. L30), $^{238}U$ (L30), and $HNO_3$ (e.g. L81). Please check this throughout the manuscript.

*Our response: Edited.*

---

## Editor Decision (ED1)

MS No.    : gchron-2024-26
Title       : The Need for Fission Track Data Transparency
Author(s)  : Murat T. Tamer et al.

**Comments to the Author:**

Thank you for submitting your manuscript to GChron. The two reviewers agreed in principle on the value of the paper, but differed somewhat in their assessment. Reviewer 2 suggested a major revision mainly because of the need to reorganize the manuscript, while reviewer 1 suggested that a minor revision is more appropriate since it is a compositional issue anyway. My opinions in this point and some other comments are as follows, and I believe that this manuscript is potentially suitable for publication in the GChron journal after minor to moderate revisions.

➢ I agree that the 6+ points mentioned by reviewer 2 are essential elements of the FT analysis, but I do not think that points not fully addressed in this manuscript need to be included in the structure of this manuscript. It would be sufficient to first state the 6+ essential ingredients of FT analysis in the introduction part (not in abstract) and then state which of them are mainly addressed in this manuscript (e.g., in L69-73). This will give the readers an overall picture of the FT analysis procedure and where this manuscript contributes or not.

➢ A common point made by the two reviewers is the recommendation to define the criteria of suitable/unsuitable/borderline grains and to use more quantitative expressions in describing the characteristics of the reported FT data. This is an important point in objectively evaluating the validity of the results/interpretations in this paper. The authors responded that they have added some important values to the main text and provided new tables to address this point. Although I cannot access the revised manuscript at this stage, I hope that this point has been carefully addressed.

➢ Reviewer 2 pointed out that etching apatite with 5M HNO3 at 20C for 20 seconds results in under-etching, but this seems to be beyond the scope of this manuscript. Since this etching recipe is widely used in FT labs around the world, this issue is too important to be discussed here. It is appropriate to provide further discussion in another paper with presenting sufficient data and evidence. The authors do not need to address the comments on this and similar issues.

In addition to the referees' comments, I will make some minor comments and technical corrections. Please consider these as well.

| | |
|---|---|
| L20-21 | I think this point is not clearly stated in the main text. The only mention of the zeta calibration in Chapter 4 is in L284-291, where the selection bias caused by differences between near-ideal standards and actual unknowns is mainly focused on, rather than improvements of the zeta method related to differences in uranium measurements and etching protocols. |
| L25 | Balestrieri et al. 1999 should be 1991 (cf. L356-358) |
| L31-32 | Hasebe et al. 2014 should be 2004 (cf. L420-421) |
| L37 | 2003a --> 2003 |
| L278 | Cogne et al. 2020 is missing from the reference list. |
| Table 1 | It would be better to include the definition of the selection rates and validity rate in the caption. Without a clear denominator and numerator, it is difficult to understand how each number is viewed. Especially for "unsuitable grain selection rate", it is difficult to know whether a larger or smaller number is better. |
| Figure 3 | The graphs are arranged horizontally, but they would be easier to read if they were arranged vertically. For example, if the graphs are arranged in order from the left column, FT density for participants 1-8, FT density for participants 9-, FT length for participants 1-8, and FT length for participants 9-, the same types of graphs are arranged vertically, which makes it easier to see comparisons between participants. |
| Reference | Green 1981 is not cited in the main body. |
| Reference | There are two "Boone et al. 2023". Distinguish between the two by adding a and b. |
| Whole text | Superscripts and subscripts do not seem to be reflected correctly in some terms, such as, $D_{par}$ (e.g. L30), $^{238}U$ (L30), and $HNO_3$ (e.g. L81). Please check this throughout the manuscript. |

---

## Author Response (AR2)

A point-by-point response to the reviewers and handling editor

**Reviewer #1**

This paper is excellent as is and should be published. I commend the Authors for this fine and most important work.

*Our Reply: We would like to express our gratitude to Dr. Donelick for his diligent efforts in preparing this review, which has significantly improved the final version of this manuscript.*

Minor edits if possible:

Lines 16-17: Close quotation on "compromising data quality and integrity"

Lines 64, 170, 366, and Table 1: Replace 37 instances of acceptable with suitable.

*Our Reply: Edited.*

**Reviewer #2**

I have read through the authors' comments on the 1st round of reviews and gone through the Author's tracked changes revised text. I think that the authors have done a vey good job of addressing the comments from the 1st round. I found numerous problems with the language usage; I have listed many suggested corrections below, but I doubt that I caught all of the problems. The co-authors are urged to go through the final version of the text.

Therefore, I think that the ms is only requires technical corrections. This will be a useful paper, particularly for teaching new trackers.

Reviewed by E. Sobel

*Our Reply: We extend our gratitude to Dr. Sobel for his constructive feedback and for dedicating his time to this review.*

show a higher percentage of selecting unsuitable grains selected a higher percentage of unsuitable grains if selected (not select), then next line should be changed to provided

*Our Reply: Edited.*

Less analysis experience analytical?

*Our Reply: Edited.*

re-analysed, a global guidance

..., this will provide a global...

*Our Reply: We changed the sentence.*

for new approaches for zeta calibration

...to zeta...?

*Our Reply: Edited.*

With the precondition of gathered from suitable grains, Six essential 'ingredients' are required for

With the prerequisite that suitable apatite crystals are available, six ....

*Our Reply: Edited.*

analyst-driven, although

Delete period

*Our Reply: We are unsure of which period the reviewer is referring to; no change made.*

participants has their had

*Our Reply: Edited.*

there has been limited ability to

...been a limited...

*Our Reply: Edited.*

underestimation of the presence of overlooked defects

This is poorly phrased. underestimation of the number of defects

How can you quantify a feature that you didn't observe?

*Our Reply: Sentence rephrased.*

73-74 may cause the analyst to lean toward "defect" for questionable features

Unclear - please rephrase

*Our Reply: Edited.*

75-6inaccurate, thereby affecting replace 'thereby' with 'and'

*Our Reply: Edited.*

both effects will bias ages lower.

... bias the ages towards lower values

*Our Reply: Edited.*

The typical number of grains for age measurements for igneous-type samples is ~10 (

I know that the other reviewer didn't like your citation for 20 grains. However you cite it, I don't think that many labs aim for 10 grains. 20 is indeed typical - just look at a handful of data tables.

*Our Reply: We have extended the text to include sedimentary rocks, for which we found another recommendation for 20 grains.*

a .xml file an, not a

*Our Reply: Edited.*

and 711 add full URL

*Our Reply: Edited.*

results we repeated results, we repeated

*Our Reply: Edited.*

Additionally, heterogeneous U distribution within the grain, judging from the distribution of spontaneous tracks, can be a complicating factor, especially if LAICPMS spot analysis is used for U determination, but also from misalignment of the spontaneous and induced track regions of interest using the EDM.

This sentence is jumbled. Judging the U distribution is more complicated in the absence of an EDM print or if the mount and print are misalignment.

*Our Reply: Sentences separated, second one rewritten.*

A confined track length is measurable as long as both ends are not exposed (Fig 1 m, n) at the surface

If an end is exposed at the surface, the track is not confined. Please rephrase.

*Our Reply: Sentence rephrased.*

delete oily. fluid, not fluids (2 fluids in 1 track is probably quite rare).

*Our Reply: Edited.*

The percentage of valid measurements is reported as the confined track length measurement validity percentage

Not a great sentence. Perhaps reported as the percentage of measured tracks/total valid tracks ? Add : see section 3.4 for details.

*Our Reply: Edited.*

Figure captions - especially Fig. 1 - when possible, please put panel labels - e.g., (a) before the relevant text rather than afterwards.

*Our Reply: Edited.*

312. Hmm. The 30% variation discussed here is so big because of one outlier analyst. Although zeta values may typically vary by 20%, there are also outliers.

*Our Reply: We have changed the sentence to place the emphasis on whether we might be asked the zeta calibration to do too much, which is the point we were getting at.*

based on the number of grains selected of each type based on the number of each grain type

*Our Reply: Edited.*

total confined track length measurements total number of …

*Our Reply: Edited.*

reviewer but by some analysts.

… but were measured by …

*Our Reply: Edited.*

not specifically how well-etched they are.

not specifically by how well etched the tracks are.

*Our Reply: Edited.*

density estimates on suitable grains

… based on … OR derived from

*Our Reply: Edited.*

density estimates on suitable of, not on

*Our Reply: Edited.*

~0.3 µm, well beyond the precision limit estimated by the standard error,

Please state this precision limit

*Our Reply: Edited.*

The participants didn't vary; their results did. How about

The confined length histograms indicate that both the number and the choice of tracks measured varied considerably between participants.

*Our Reply: Edited.*

measurements on suitable grains from OR of Same comment 357

*Our Reply: Edited.*

provide have

*Our Reply: Edited.*

Table 3 caption the number of grains selected of each type the number of grains of each type selected,

*Our Reply: Edited.*

Table 4 caption the total confined track length measurements for each analyst the total number of confined track length measurements for each analyst

*Our Reply: Edited.*

but number but the number

*Our Reply: Edited.*

the track density increases by ~35%.

increased

*Our Reply: Edited.*

Excluded defect areas and tracks.

This sentence fragment is missing some words. What are the green tracks? This should be stated in the caption - ah - the missing words!

*Our Reply: Edited.*

delete 'to'

*Our Reply: Edited.*

delete 'for'. Change 'against' to 'using

*Our Reply: Edited*

perhaps you should add a reminder: "The analyst should remember that not all samples yield usable data."

*Our Reply: Edited.*

Perhaps note something like "However, it is presently unclear how such data can be modelled with the existing annealing equations.

*Our Reply: We added a statement to this effect.*

"668 change 'to avoid' to 'that should be avoided'

*Our Reply: Edited.*

Exposure is a poor word. Publication.

*Our Reply: Edited.*

change 'to' to 'in'

*Our Reply: Edited.*

also pathways also presents pathways

*Our Reply: Edited.*

beginning of the fission-track dating method fission-track dating method was first established

*Our Reply: Edited.*

Here's a question - is it possible for people to not just look at the grains in this study but also to analyze them in the same way as was done in this study? I.e. - to use this study as a teaching tool.

*Our Reply: Someone could easily use this image set as a teaching tool, but since we did not design it as one, we do not necessarily want to put it forward as one.*

We appreciate all the participants appreciate the work (or some other word) of all of the participants.

*Our Reply: Edited.*

yes, the copy editor will fix this, but references should be in alphabetical order.

*Our Reply: Edited.*

**Handling editor**

Two reviewers confirmed that the previous review comments were adequately addressed in the revised manuscript. From a scientific point of view, I judge that the manuscript has reached a level worthy of publication in the GChron journal. The reviewers have suggested some additional corrections to the wording of the text, and I would like to ask the authors to address these technical corrections before the publication. Also, of the previous AE comments, the comment on Figure 3 was not addressed, so please consider this as well, if possible.

*Our Reply: We thank the editor for the comments and handling the manuscript. The histograms in Fig 3 are arranged accordingly.*